# RNA language models predict mutations that improve RNA function

Yekaterina Shulgina [1,2,3,12], Marena I. Trinidad [1,4,12], Conner J. Langeberg [1,2,3,12], Hunter Nisonoff [5,12], Seyone Chithrananda [1,6,12], Petr Skopintsev [1,3,12], Amos J. Nissley [7,12], Jaymin Patel [1], Ron S. Boger [1,8], Honglue Shi [1,4], Peter H. Yoon [1,2], Erin E. Doherty [1,3], Tara Pande [6], Aditya M. Iyer [9], Jennifer A. Doudna [1,2,3,4,7,10,11] & Jamie H. D. Cate [1,2,3,7,10,12] ✉

Structured RNA lies at the heart of many central biological processes, from gene expression to catalysis. RNA structure prediction is not yet possible due to a lack of high-quality reference data associated with organismal phenotypes that could inform RNA function. We present GARNET (Gtdb Acquired RNa with Environmental Temperatures), a new database for RNA structural and functional analysis anchored to the Genome Taxonomy Database (GTDB). GARNET links RNA sequences to experimental and predicted optimal growth temperatures of GTDB reference organisms. Using GARNET, we develop sequence- and structure-aware RNA generative models, with overlapping triplet tokenization providing optimal encoding for a GPT-like model. Leveraging hyperthermophilic RNAs in GARNET and these RNA generative models, we identify mutations in ribosomal RNA that confer increased thermostability to the *Escherichia coli* ribosome. The GTDB-derived data and deep learning models presented here provide a foundation for understanding the connections between RNA sequence, structure, and function.

RNAs serve many fundamental roles in biology ranging from gene expression to catalysis, and can adopt complex three-dimensional folds to carry out these functions. Inspired by the successes in protein structure prediction[1,2], multiple groups have made progress towards developing deep learning models for RNA secondary and tertiary structure prediction[3–10]. However, based on assessment of the CASP15 RNA modeling challenge and the metrics used therein, RNA structure prediction using deep learning approaches has not reached human-tailored model performance, and human modeling of RNA structure is still not at the level of protein structure prediction[11–14]. A fundamental

weakness in RNA modeling is the state of RNA sequence, structural, and phenotypic databases available for training deep learning models[13,15]. Rfam, the closest analogue to Pfam for proteins[16], provides curated seed sequences, alignments and homology models for thousands of RNA families[17]. However, Rfam alignments have limited phylogenetic scope, only drawing from Uniprot reference genomes ($n = 14,451$). The SILVA database contains highly-curated information for 16S and 23S ribosomal RNA (rRNA) sequences[18], but not for other RNAs. Another major database, RNAcentral, aggregates RNA sequence and structural information from a range of RNA databases[19]. However,

[1]Innovative Genomics Institute, University of California, Berkeley, CA, USA. [2]Department of Molecular and Cell Biology, University of California, Berkeley, CA, USA. [3]California Institute for Quantitative Biosciences, University of California, Berkeley, CA, USA. [4]Howard Hughes Medical Institute, University of California, Berkeley, CA, USA. [5]Center for Computational Biology, University of California, Berkeley, CA, USA. [6]Department of Electrical Engineering and Computer Sciences, University of California, Berkeley, CA, USA. [7]Department of Chemistry, University of California, Berkeley, CA, USA. [8]Biophysics Graduate Program, University of California, Berkeley, CA, USA. [9]Department of Physics, University of California, Berkeley, CA, USA. [10]MBIB Division, Lawrence Berkeley National Laboratory, Berkeley, CA, USA. [11]Gladstone Institutes, University of California, San Francisco, CA, USA. [12]These authors contributed equally: Yekaterina Shulgina, Marena I. Trinidad, Conner J. Langeberg, Hunter Nisonoff, Seyone Chithrananda, Petr Skopintsev, Amos J. Nissley, Jamie H. D. Cate. ✉e-mail: j-h-doudna-cate@berkeley.edu

RNAcentral overrepresents rRNAs, tRNAs, lncRNAs and a few small RNA families (i.e. snRNAs, snoRNAs, miRNAs, and piRNAs). Furthermore, some of the underlying databases are no longer maintained or updated, or have substantial sequence overlap leading to redundant entries in the database. Taken together these databases are far less extensive than protein databases that include hundreds of millions of unique sequences[16]. Furthermore, a related fundamental challenge with RNA structure prediction is the difficulty in building robust sequence alignments for intact functional RNAs due to limitations in identifying their 5' and 3' ends and the uneven sampling of sequences across phylogeny[20,21]. Finally, the number of available high-quality RNA structures in the Protein Data Bank (PDB)[22] lags those for proteins by orders of magnitude, and is heavily biased towards a small number of RNA structural types, particularly those found in ribosomes[13].

The ribosome is a major target for engineering an expanded genetic code[23]. Ribosomal RNA (rRNA) catalyzes peptide bond formation by the ribosome, and many efforts have attempted to use directed evolution of rRNA to engineer ribosomes that can incorporate non-proteinogenic monomers into polypeptides[24–27]. However, the complexity of ribosome assembly constrains directed evolution of the ribosome for novel functions[28–31]. In *Escherichia coli*, ribosomes comprise three ribosomal RNAs (5S, 16S, and 23S rRNAs) and 54 proteins, along with many protein factors required to assemble them in cells. As a result of this complexity, ribosomes obtained from directed evolution experiments often have defects in their assembly and lose activity[32]. Strategies for the directed evolution of proteins for new function often begin with thermostable proteins, which are more robust to mutations required to recover functional variants[33–35]. It is presently infeasible, however, to replace the *E. coli* large ribosomal subunit RNA (23S rRNA) with a thermostable 23S rRNA from another organism, as efforts at 23S rRNA directed evolution beginning from the rRNA from other organisms have so far been unsuccessful[29].

Here we leveraged the Genome Taxonomy Database (GTDB)[36] to build more comprehensive RNA sequence databases and alignments. The GTDB provides a standardized taxonomy across all high-quality bacterial and archaeal genomes including metagenome-assembled genomes and single-cell amplified genomes. This greatly expands the available RNA sequence diversity, as the vast majority of microbes are unculturable. Furthermore, as a standardized taxonomy, the GTDB provides a framework for linking sequence data to phenotypes and other experimental data, which are often limited to cultured microbes. The taxonomy presently includes over 400,000 bacterial and archaeal genomes organized around over 85,000 species clusters, which provides a rich resource for principled genomic comparisons, sequence analysis and sequence alignment. We find that RNA sequences mined from GTDB genomes represent a more diverse set of sequences than state-of-the-art databases with only one clear exception–16S rRNA. We mapped growth temperatures from other sources to the GTDB and used an existing machine-learning approach to predict optimal growth temperatures for reference genomes lacking direct growth temperature information. We combined these with the RNA sequences mined from the GTDB to create the GARNET (Gtdb Acquired RNa with Environmental Temperatures) database. Using GARNET, we developed two types of machine-learning models to map sequences to functional properties of the RNA. We trained a compact RNA generative Graph Neural Network (GNN) using a 23S rRNA multiple sequence alignment (MSA) with structural conditioning. We also trained Generative Pretrained Transformer (GPT)-like RNA language models that revealed an optimal triplet encoding for RNA. By finetuning these RNA generative models on hyperthermophilic RNA sequences, we were able to predict mutations in the *Escherichia coli* ribosome that increased its thermostability. These results open new approaches to expand computational algorithms for predicting RNA structure and altering RNA function in biology.

## Results

### Building RNA sequence datasets from GTDB genomes

To generate diverse and minimally-redundant alignments of RNA sequence families for the GARNET database, we turned to the GTDB genomes which represent 80,789 bacterial and 4416 archaeal species clusters (release 214.1) (Fig. 1a). First, we built an rRNA sequence dataset by searching each GTDB species reference genome for 23S, 16S, and 5S rRNA sequences. Searches were performed with Infernal[20] using the corresponding Rfam covariance models (CMs), taking the top hit per genome with an e-value < 1e-5 and aligning to at least 85% of the consensus CM sequence. If no such hit could be found, we additionally searched the available non-representative genomes in each species cluster. We further ensured alignment quality by removing hits that broke a substantial fraction of the consensus base pairs or had exceedingly long insertions (see Methods for details). For 23S rRNA, which is roughly 2.9 kb in length, we identified a 23S rRNA sequence for 32,317 species (Fig. 1b). The absence of a full-length 23S or 16S rRNA sequence in many genomes likely reflects the fragmented nature of some metagenome-assembled genomes and the occasional presence of introns that cause partial hits. We additionally searched all GTDB representative genomes for 228 RNA families using Rfam models that are likely to occur in bacteria or archaea and are over 100 nucleotides long, applying the same quality-control criteria as for ribosomal RNAs except allowing for multiple hits per genome. This search identified a total of 714,662 sequences, with the seven largest families comprising 58% of the 228 RNA sequence dataset (Fig. 1c).

We evaluated the sequence diversity of the GTDB-derived datasets by assessing the number of unique sequences at different fractional identity thresholds compared to state-of-the-art datasets for these RNA families. For 23S and 16S rRNA alignments, we compared against the SILVA database[18]; for 5S rRNA, we compared against the 5SRNAdb[37] and the Rfam full alignment[17]; for the top three most abundant of the 228 RNA families (T-box leader, cobalamin riboswitch, and TPP riboswitch), we compared against Rfam full alignments. In all cases, except for 16S rRNA and 23S rRNA, the GTDB-derived alignments had substantially greater sequence diversity compared to the state-of-the-art dataset (Fig. 1d, e, Supplementary Fig. 1c, d). For 23S rRNA, the SILVA database had comparable diversity to the GTDB-derived alignment, and for 16S rRNA, the SILVA database had greater diversity (Fig. 1d, Supplementary Fig. 1c), likely due to the widespread use of 16S rDNA sequencing of new microbial isolates and environmental samples. Taken together, these results highlight the benefit of using the GTDB as a framework for building comprehensive RNA sequence datasets.

### Mapping optimal growth temperatures to GTDB reference genomes

The GTDB taxonomic framework allows us to link RNA sequences derived from the GTDB genomes to phenotypes, which can aid in RNA modeling and engineering. We chose to map GTDB species to optimal growth temperatures (OGTs) from TEMPURA[38] and Gosha[39] databases. However, since the TEMPURA and Gosha databases only include cultivated species, they only have experimental OGTs for 15% of the GTDB reference species. We therefore inferred OGTs of all GTDB reference genomes using TOME[40]. TOME predicts the OGT for an organism using a machine learning model trained on proteome-wide dipeptide (2-mer) distributions. Importantly, TOME was trained on only a subset of organisms now available in the TEMPURA and Gosha databases. We therefore used these new organisms to validate TOME predictions, and found that the predicted OGTs correlated well with the TEMPURA and Gosha sets not used for TOME training (Fig. 2a, Supplementary Data 2; $R^2$ values of 0.868 and 0.881, respectively). We also used isolation source metadata associated with each GTDB reference genome as a check on TOME OGT

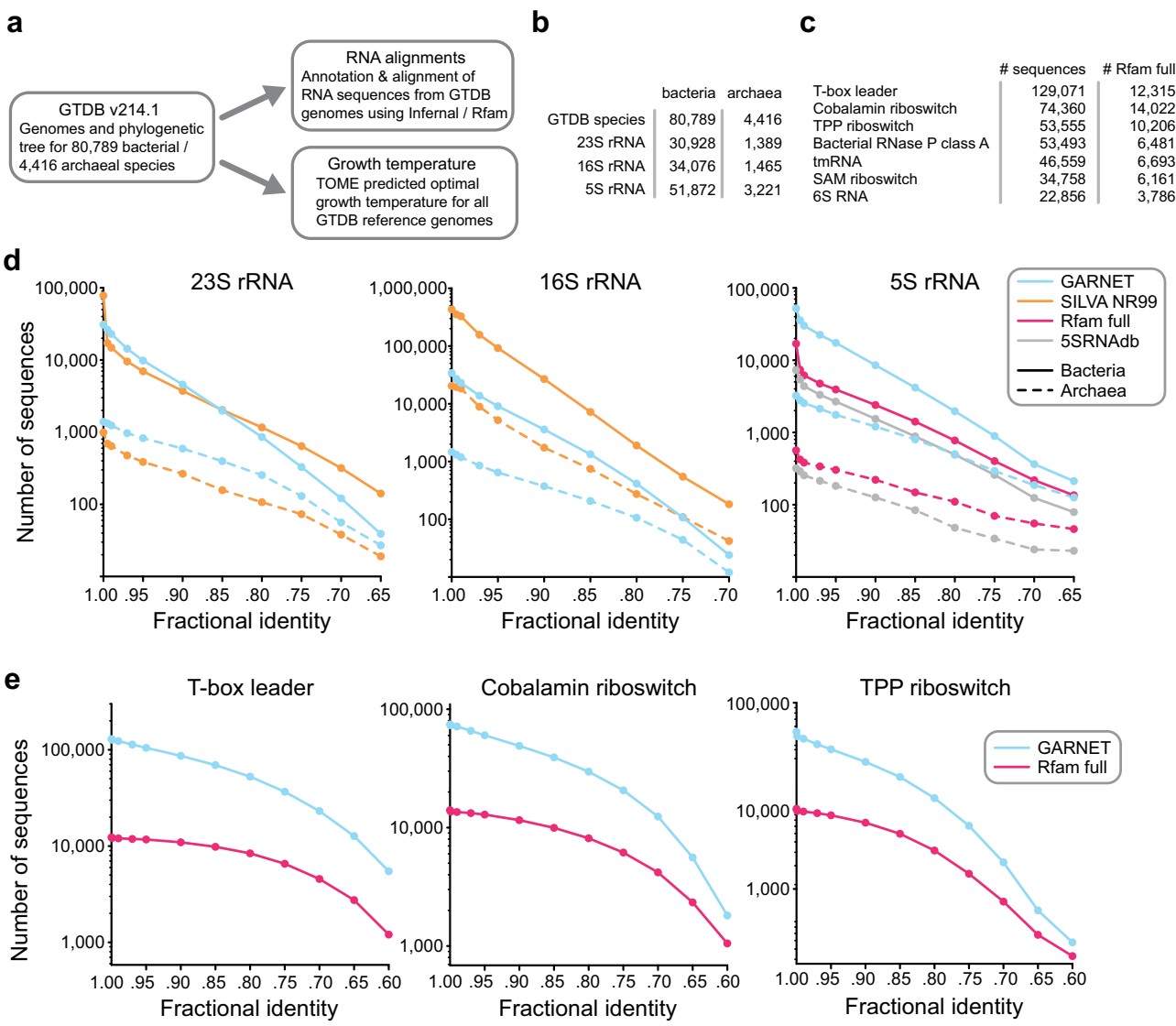

**Fig. 1 | The Genome Taxonomy Database as a source for RNA sequences.**
**a** Construction of the GARNET database centered on the GTDB structure, linking RNA alignments mined from GTDB genomes with growth temperature prediction through a consistent taxonomy. **b** Number of GTDB species found to have at least one high-quality, near-full length hit for 23S, 16S, and 5S rRNA. **c** Top seven non-rRNA Rfam families with most sequences found in GTDB representative genomes compared against the Rfam full alignment. In contrast to the rRNA alignments, multiple sequences per genome were allowed. Information for the entire 228 RNA dataset can be found in Supplementary Data 1. **d** Comparing diversity of GARNET RNA sequences against state-of-the-art datasets for 23S rRNA, 16S rRNA, and 5S rRNA by filtering the sequences at a range of pairwise fractional identity thresholds with VSEARCH[58]. **e** Diversity comparison for the top three most abundant of the 228 RNA families in GARNET with VSEARCH.

predictions, especially for uncultivated species. Although the isolation source of each organism is heterogeneous in terminology and may not reflect the actual optimal growth conditions, we found that the metadata with unambiguous source information is consistent with TEMPURA and Gosha OGTs (Fig. 2b, Supplementary Data 2) (See Methods). This is also true for OGTs predicted using TOME (Fig. 2b, Supplementary Data 2). Interestingly, TOME predicted hyperthermophilic species (OGT > = 60 °C) in both archaea and bacteria in clades with no known hyperthermophiles in the TEMPURA or Gosha databases (Fig. 2c, Supplementary Fig. 2, Supplementary Data 2). These results provide a rich resource for inferring the physiological temperature at which RNAs and proteins from GTDB organisms function optimally. We combined the GTDB-derived RNA sequences with the TOME-predicted OGTs to create the GARNET (Gtdb Acquired RNa with Environmental Temperatures) database, to use for training new RNA deep learning models.

## A sequence and structure based RNA generative model for 23S rRNA

Generative deep learning models that integrate structural information provide highly compact representations of protein families that have proven useful for protein design[41]. These models leverage the fact that structure is generally conserved within protein families. We extended this framework to RNA, creating compact structure-informed models to circumvent scalability constraints inherent to the extensive length of 23S RNA. We harnessed the sequence diversity within the GTDB and the wealth of high-resolution structures available for the large ribosomal (50S) subunit to develop a Graph Neural Network (GNN) model. For 23S rRNA, the known representative 3D structures provide abundant information to benchmark MSAs and better model the RNA family. Our generative model inputs a distance matrix for the representative structure of the family[42], and is trained on next-token prediction for an aligned MSA[41].

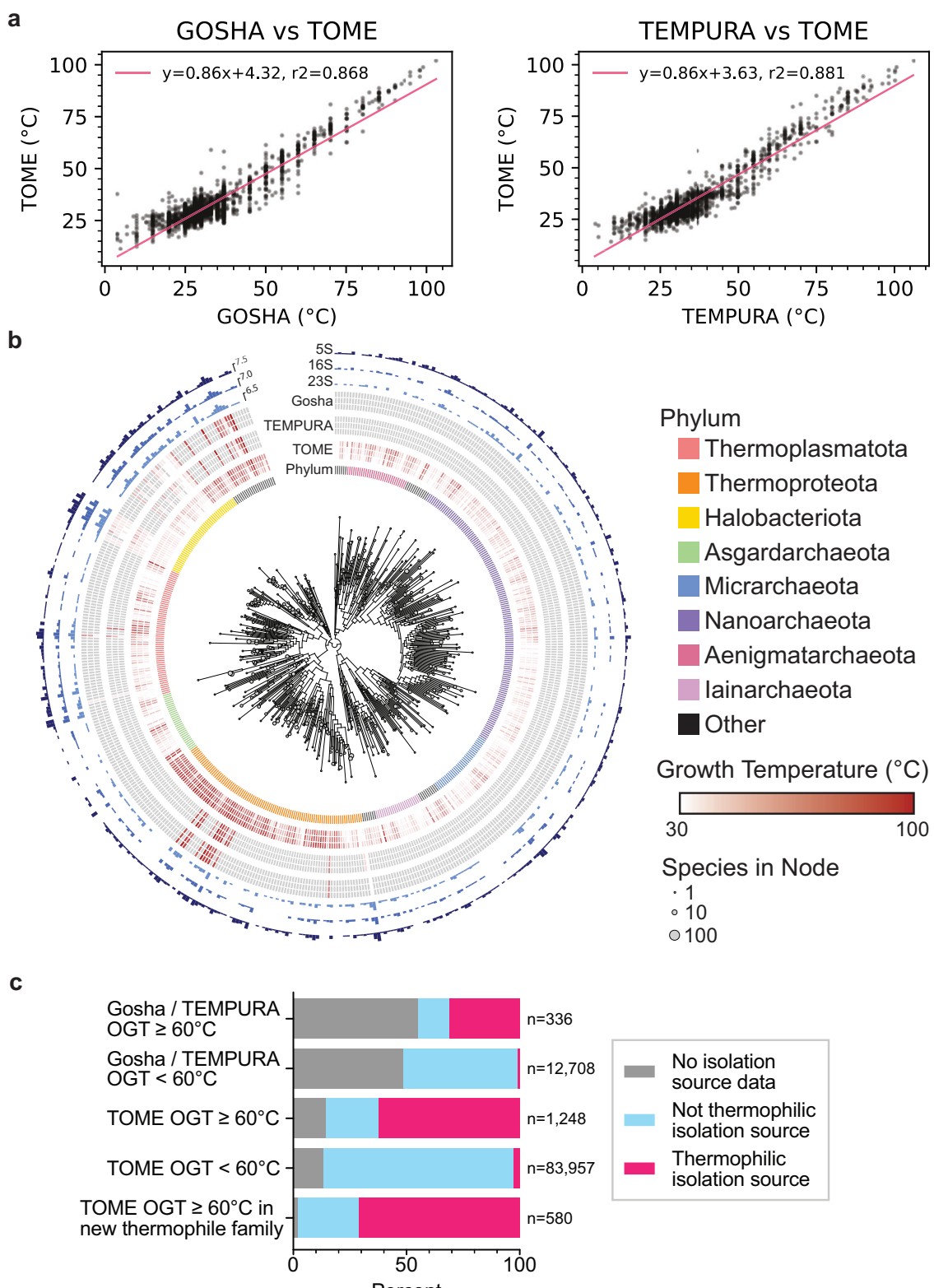

**Fig. 2 | Optimal growth temperatures of GTDB reference organisms.**
**a** Correlation of TOME-predicted and experimental OGTs from Gosha and TEM-PURA, excluding species from TOME's training set ($n = 3346$ and $7404$ species, respectively). **b** Archaeal phylogenetic tree of GTDB reference organisms, grouped at the Family taxonomic rank, arbitrarily rooted. A similar tree for bacteria is in Supplementary Fig. 2. Node tip sizes are proportional to the number of species represented by node ($\log_2$ transformed). Inner circle indicates Phylum. The next circle represents TOME-predicted min, median, and maximal optimal growth temperatures of all species within rank. The next two circles similarly represent empirically measured optimal growth temperatures pulled from the TEMPURA and Gosha datasets, respectively. Outer circles represent the total number of 23S, 16S, and 5S detected in each rank, respectively ($\log_2$ transformed). **c** Thermal isolation sources for GTDB bacterial and archaeal species (manually classified from GTDB metadata) comparing species with hyperthermophilic OGTs ($\geq 60\,^{\circ}\mathrm{C}$) in the Gosha / TEMPURA databases, TOME hyperthermophiles, and non-hyperthermophiles. The bottom bar corroborates TOME hyperthermophiles ($n = 580$) with no close hyperthermophilic relatives in Gosha / TEMPURA (family-level).

The model leverages a graph-based representation of the RNA structure to build a sparse attention mechanism (i.e. Graph Attention Network) in which the positions attend to their $k$-nearest neighbors in structure space at each layer (Fig. 3a, b). We pre-processed the 23S rRNA MSA of the GARNET sequences for training. The corresponding graph was created by choosing $k$-nearest neighbors to each nucleotide from a distance matrix of *E. coli* 23S rRNA, aligned with the MSA so that nucleotides in matrix columns and rows match their counterparts in the MSA (see Methods). This matrix was derived by calculating the minimal interatomic distances between nucleotides pairs in the 23S rRNA. We found that using $k = 50$ nearest neighbors provided an optimally trained model, with respect to model size and perplexity (Fig. 3c). For the model input analysis, the distance matrix was transformed into a binary contact map by selecting the $k$-nearest neighbors for each nucleotide (see Supplementary Fig. 2). We found that at $k = 50$ nearest neighbors, the model samples all contacts below ~12 Å, and a subset of longer-range contacts up to 24 Å, or distances at which inter-helical packing can be detected (see Fig. 3d–f, Supplementary Figs. 3 and 4). Complete model specifications are available in Supplementary Data 3.

## A modified GPT language model for RNA

AlphaFold relies on MSAs as a central component of an end-to-end deep learning algorithm for protein structure prediction[1]. However, large language models for proteins such as ESM-2 replace MSAs in structure prediction, and are particularly useful when MSA information is lacking. In the case of RNA, obtaining robust MSAs can be challenging[20,21], even with databases as large and diverse as GARNET. Furthermore, whereas GNNs require a structural prior for training, language models are not restricted by structural constraints or assumptions about RNA flexibility or whether an RNA might adopt multiple folds. We therefore tested whether a language model (LM) for RNA could be developed using sequences from GARNET. We first modified a compact GPT model architecture–nanoGPT[43]– for training on RNA sequences and tested different methods of tokenizing nucleotides (Fig. 4a). Using 23S ribosomal RNA (rRNA) sequences from GARNET (Fig. 1, Supplementary Data 1), we found that models trained using tokens representing three nucleotides, with a 1-nucleotide shift per token, performed substantially better than using either individual nucleotides or paired nucleotides (Fig. 4b, Supplementary Data 3 and Methods). We also found using rotary positional embedding (RoPE)[44] in each attention layer allowed RNA LMs to be trained with paired-nucleotide encodings. However, paired-nucleotide tokenization required training models with a slower learning rate, and these models had a higher validation perplexity than models using RoPE with triplet-nucleotide encoding (Fig. 4b). In addition to 23S rRNA, we also trained a more general RNA LM using sequences from 231 RNA families in GARNET (228 RNA dataset plus three rRNA datasets), as described above (Fig. 4c). These models had lower validation perplexities compared to the RNA LMs trained only on 23S rRNA sequences (Supplementary Data 3). They also are capable of generating RNA sequences that align with full-length 23S and 16S rRNA when queried with their respective 5′ ends (Fig. 4d).

## Finetuning RNA generative models with hyperthermophilic sequences

Replacing the *E. coli* 23S rRNA with a thermostable 23S rRNA from another organism is presently not feasible[29]. We therefore tested whether finetuning the GNN and RNA LM models using hyperthermophilic 23S rRNAs could help identify mutations that make the *E. coli* ribosome more stable for future directed evolution efforts. We finetuned the GNN and RNA LM pretrained models described above using 23S rRNA sequences from hyperthermophilic bacteria and archaea with TOME-predicted OGTs of 60°C or higher (Methods). We then used the resulting pretrained and finetuned models to generate sets of

1000 RNA sequences seeded with the 5′-end of *E. coli* 23S rRNA, and a range of "temperature" scaling factors to modulate the probabilities of token generation (Methods).

We assessed the quality of the RNA sequences generated from the models, i.e. how "23S-like" they are, by comparing them to the covariance model for bacterial 23S rRNA in Rfam (RF02451) using cmsearch in the Infernal suite of programs[20,29]. We evaluated the full set of 23S rRNA sequences in the GARNET database as a control. Naturally occurring sequences in GARNET had cmsearch scores that clustered around 1900 and 2700 for archaeal 23S and bacterial 23S, respectively (Fig. 5a-d). Sequences generated from the GNN had high cmsearch scores within the range of natural sequences, although these dropped at higher generation temperatures likely due to the dropout of local RNA sequence segments (Fig. 5a, b, and Supplementary Fig. 5). Sequences generated by the RNA LMs also had high cmsearch scores, suggesting they have bacterial 23S rRNA-like properties across all generation temperatures tested (Fig. 5c, d). At lower generation temperatures, the finetuned RNA LM generated some sequences that harbored long stretches of repetitive sequence, resulting in low cmsearch scores (Supplementary Fig. 6c, d).

We also examined secondary structure preservation as a separate measure of the 23S-like properties of the generated sequences. Naturally occurring 23S rRNAs typically contain a small percentage of non-canonical base pairs (i.e. base pairs other than standard Watson-Crick-Franklin and G-U pairs) in the consensus secondary structure for RF02451 model (Fig. 5e–h). Sequences generated by the pretrained RNA LM retained a similar proportion of non-canonical base pairs up to a generation temperature of 0.9, while the finetuned models inserted more non-canonical pairs relative to natural sequences at temperatures higher than 0.5 (Fig. 5g, h, and Supplementary Fig. 6d). The GNN models started to include a higher percentage of non-canonical pairs at generation temperatures of 0.6 or higher (Supplementary Fig. 5d). Taken together, these quality control measures inform selection of sequence generation temperatures that can aid subsequent analyses of sequences generated from the 23S rRNA GNNs and RNA LMs trained on GARNET sequences (Supplementary Fig. 7, Supplementary Data 3).

## Sequences generated by GenerRNA

Separately, we attempted to use a different generative RNA language model implemented based on the nanoGPT code[43,45]. This implementation of nanoGPT, called GenerRNA, was pretrained using the RNAcentral database[19], and used a byte pair encoding (BPE) algorithm[46] to generate a 1024 token library for RNA. We used the GenerRNA model pretrained on RNAcentral sequences, and also finetuned this model on the GARNET-all and GARNET-hyperthermophile sequences using the provided GenerRNA tokenization and training code. We then generated sets of 1000 23S sequences, analogous to the process for the RNA LMs. However, none of the three GenerRNA models was capable of generating full-length 23S-like sequences (Supplementary Fig. 8, see Methods).

## Identifying mutations to stabilize the *E. coli* ribosome

To identify potential mutations to the *E. coli* 23S rRNA that might confer thermostability, we examined sequences generated from the 23S rRNA GNN and LM pretrained and finetuned models (PT and FT models, respectively) using a generation temperature of T = 0.5 (Methods). We first compared the Jensen-Shannon divergence (JSD) of nucleotide frequency distributions of the FT-generated sequences relative to the PT-generated sequences, after masking the positions used as the seed as well as those with less than 50% occupancy in the alignment (Supplementary Data 4). We also calculated the JSD of natural hyperthermophilic 23S rRNA sequences used for finetuning relative to the entire GARNET 23S rRNA set (Supplementary Data 4). 23S rRNA positions with high JSDs differ the most in which nucleotides are generated by the PT and FT models, indicating mutations that may be

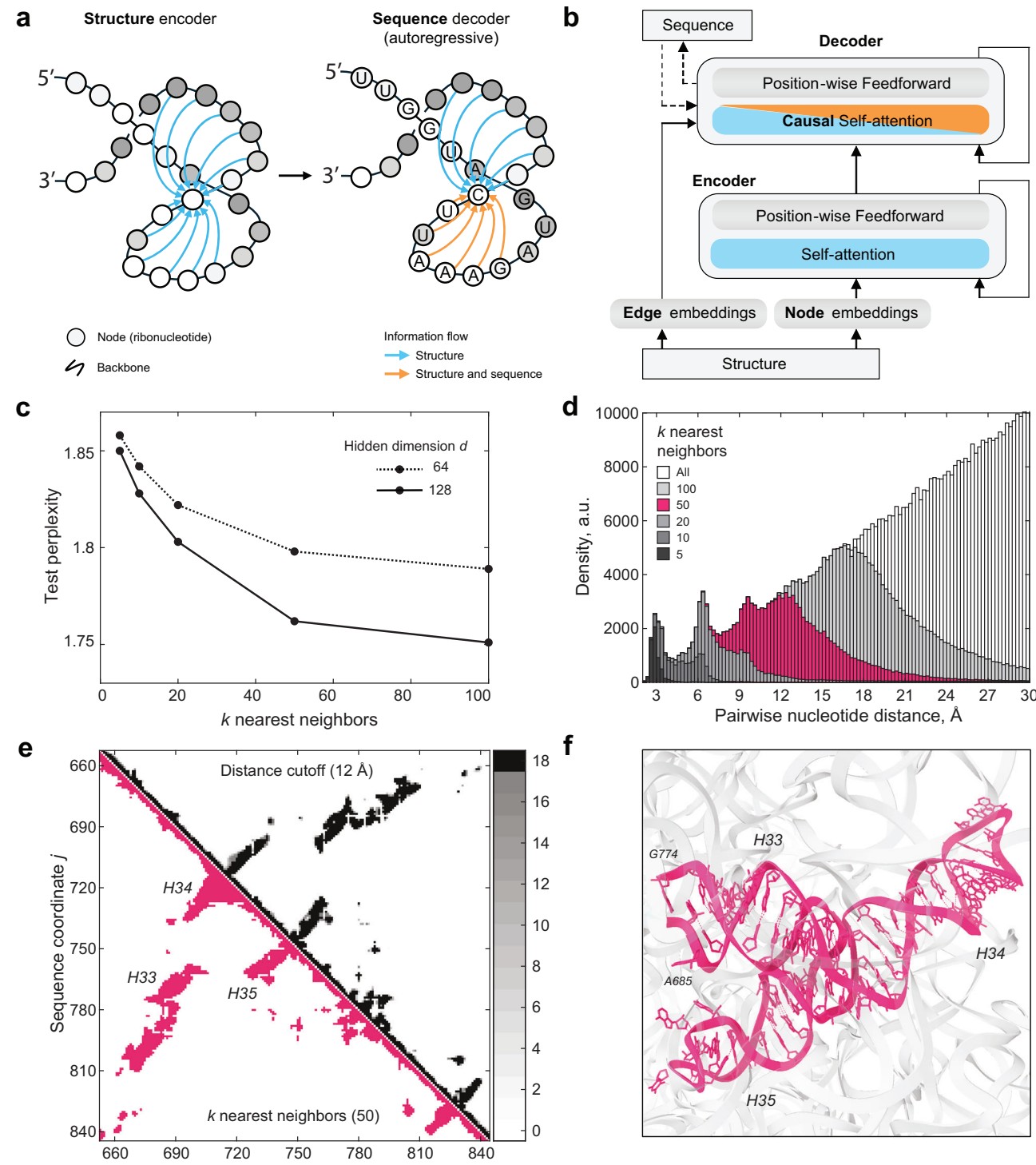

**Fig. 3 | A 23S rRNA generative model using GTDB sequences and large ribosomal subunit structures. a** Graph Neural Network (GNN) model schematic. **b** GNN model architecture. Panels (**a**) and (**b**) are adapted from Ingraham et al[41]. to illustrate their use for RNA. For detailed model parameters and training data statistics refer to Supplementary Data 3. **c** Test perplexity of the GNN models plotted as a function of $k$-nearest neighbors, highlighting that the model does not significantly improve for $k$ values greater than 50. The final perplexity of the model with hidden dimensions $d = 128$, and $k = 50$ was 1.751. **d** Histogram of internucleotide distances sampled by selecting $k$ nearest neighbors in the distance matrix for *E.coli* 23S rRNA structure (PDB ID: 7K00)[42]. Choosing $k = 50$ covers all

distances less than 12 Å. **e** Comparison of the contact maps generated from the distance matrices, based either on the distance cutoff or the $k$ nearest-neighbors criteria (see Methods). Top-right, the sum of the contact maps for 18 bacterial and archaeal ribosomal RNA structures, projected onto the MSA sequence alignment, and based on the 12 Å distance cutoff criterion. The number of contact maps that align for a given pair of nucleotides is color-coded in the color bar on the right. Bottom-left, contact map for *E. coli* 23S rRNA, based on selecting $k = 50$ nearest neighbors to each nucleotide. The two types of contact maps show high similarity. **f** Structure of the three stem-loops highlighted in (**e**). A 12 Å inter-helical packing contact is shown with a dashed line in (**f**), and with an arrow in (**e**).

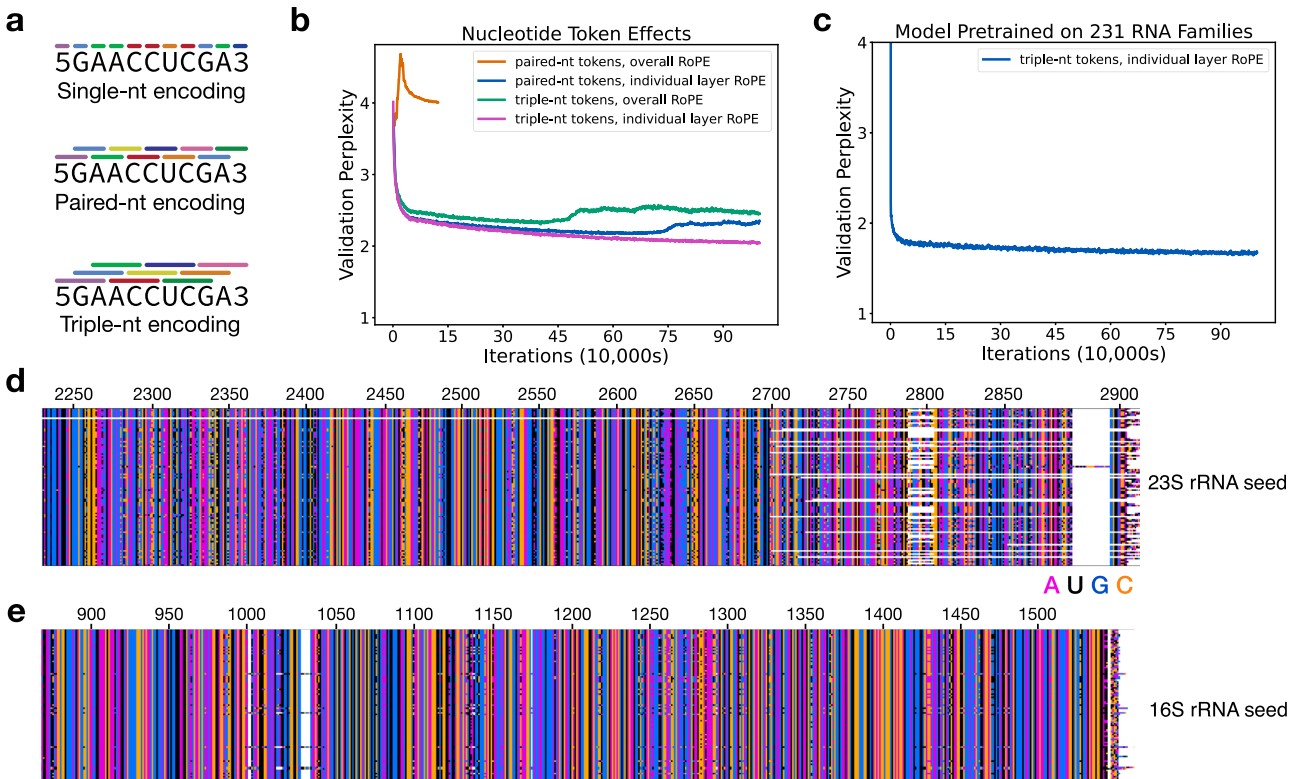

**Fig. 4 | Tokenization schemes for RNA language models. a** Representation of nucleotides as tokens for single, paired, or triplet nucleotides. Tokens are encoded for nucleotides in 1-nucleotide steps, i.e. are overlapping for paired and triplet nucleotides. Beginning and end tokens are also included in the token library.
**b** Perplexity of RNA language models trained on 23S rRNA sequences, with the nanoGPT model modified to use an overall rotary positional embedding (RoPE), or with RoPE applied to each attention layer. Training with paired-nt and overall RoPE was conducted for 100,000 iterations, whereas the other models were trained for 1 M iterations, with a batch size of 18 in all models. A perplexity value of 4 would be random (i.e. 4 nucleotides to choose from), and a value of 1 would indicate perfect certainty in nucleotide choice. The perplexity after training for a random model should be 4 regardless of the tokenization scheme, due to the 1-nucleotide steps used with the paired and triplet encoding. **c** Perplexity of an RNA LM pretrained on 231 RNA sequence families in GARNET (Supplementary Data 1). The perplexity of an RNA LM model finetuned on hyperthermophilic RNAs, starting from the pretrained general model, is 1.33. For detailed model parameters and training data statistics in panels (b) and (c) refer to Supplementary Data 3. **d** Alignment of 23S rRNA sequences generated using the more general pretrained 231-RNA LM, showing the 3' end of the generated sequences ($n = 100$). **e** Alignment of 16S rRNA sequences generated using the more general pretrained 231-RNA LM, showing the 3' end of the generated sequences ($n = 100$). Sequence generation in panels (**d**) and (**e**) was seeded with 100 nucleotides of *E. coli* 23S rRNA or 16S rRNA, respectively, and using a temperature of 0.2. The bottom row is the *E. coli* sequence, and *E. coli* nucleotide numbering is also shown. White space shows regions where insertions and deletions are present in the sequences.

important for thermostability (Supplementary Figs. 9–11). Interestingly, there was very little overlap in positions with the highest JSDs when comparing GNN- or RNA LM-generated sequences to those predicted by comparing natural sequences, whereas there was substantial overlap between the deep learning approaches (Fig. 6a). Nucleotide positions predicted to confer thermostability using the 231-RNA trained LMs also differed from those obtained from natural sequences in GARNET (Fig. 6a). These results show that the deep learning models predict nucleotide changes in *E. coli* 23S rRNA that differ markedly from those that could be gleaned from the 23S rRNA data in GARNET.

Although sorting by JSD can help identify candidate stabilizing mutations, individual mutations may depend on sequence context and may require evaluation as part of an entire 23S rRNA sequence. Furthermore, the generated sequences may not represent all stabilizing mutations learned by the models. For example, a rare sequence variation in the A loop of 23S rRNA at positions U2554 and U2555 only occurs in a single phylum of archaeal hyperthermophiles, Thermoproteota, in which one or both nucleotides are mutated to a C[47]. These mutations in the *E. coli* ribosome are known to improve ribosome stability[47], yet neither position appears as a top candidate using the JSD filtering described above. To assess whether the GNN and RNA LM FT models support these mutations, we calculated the probability of

generating mutant *E. coli* 23S rRNA sequences. Since the models were trained on sequences similar to *E. coli*, mutations away from the wildtype (WT) *E. coli* sequence often lead to lower probabilities. We therefore compared the probability of generating a mutant *E. coli* 23S sequence from the FT model relative to the PT model, and normalized it to that of the WT sequence ($\Delta\Delta logP$) (Fig. 6b). Using this methodology, a U2554C mutation is supported by the FT model better than 85.4% and 72.3% of all possible single mutants when evaluated by the 23S LM and 231-RNA LM, respectively, and 57.4% of single mutants when evaluated by the GNN model (Supplementary Fig. 12 and Supplementary Data 5), consistent with the moderate increase in thermostability seen with *E. coli* 50S subunits harboring this mutation[47]. We also found that the combined U2554C-U2555C mutation had a positive $\Delta\Delta logP$ predicted from the GNN and RNA LMs (Supplementary Data 5). Taken together, JSD-based sorting and the use of model probabilities help identify sites in 23S rRNA that could confer higher thermostability to the *E. coli* 50S subunit.

**Testing 23S rRNA mutations predicted to stabilize the ribosome**
One of the strongest predictions from the LM and GNN models for a mutation that could confer thermostability to the *E. coli* 50S subunit occurs in the closing loop at the end of helix H89 in 23S rRNA, adjacent to the peptidyl transferase center of the ribosome. The H89 stem-loop

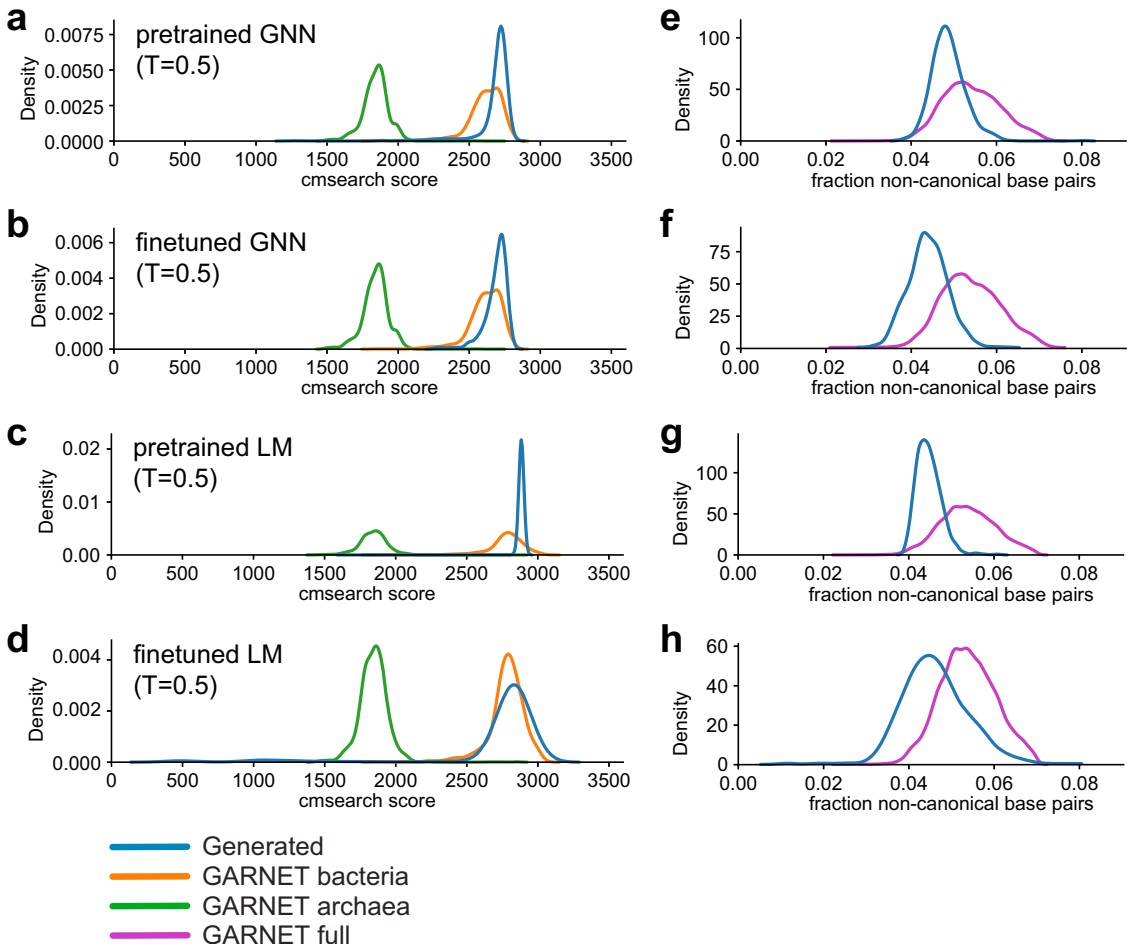

Generated
GARNET bacteria
GARNET archaea
GARNET full

**Fig. 5 | 23S rRNA sequences generated by GNN and GPT-like RNA models.**
**a**–**d** Cmsearch scores for sequences generated from the pretrained GNN model (**a**),
finetuned GNN model (**b**), pretrained RNA LM (**c**), and finetuned RNA LM (**d**) trained
on 23S rRNA sequences at generation temperature T = 0.5 compared to naturally
occurring 23S rRNAs in GARNET. For the GARNET reference distributions, random
subsets of 1000 bacterial sequences and 1000 archaeal sequences were used.

**e**–**h** 23S rRNA sequences generated from the pretrained GNN model (**e**), finetuned
GNN model (**f**), pretrained RNA LM (**g**), and finetuned RNA LM (**h**) according to the
fraction of disrupted canonical base pairs (i.e. Watson-Crick-Franklin and G-U)
relative to the Rfam RF02541 consensus secondary structure (denoted non-
canonical base pairs) in the generated sequences compared to naturally-occuring
23S rRNAs.

folds late in 50S subunit assembly and also engages with ribosome
assembly factors[48,49]. We, therefore, examined the JSDs of generated
sequences and model probabilities in this region for potential muta-
tions that might stabilize the ribosome. The finetuned GNN model and
both finetuned LMs predict a U to C mutation in the apical loop of H89
at position 2477 to confer thermotolerance using the JSD calculation
(ranked 65, 18, or 178 by the 23S rRNA GNN, 23S rRNA LM, or 231-RNA
LM, respectively). By contrast, nucleotide 2477 is not a top hit when
using the JSD metric on natural GTDB sequences (ranked 1007 out of
2904 positions). Introducing the U2477C mutation in *E. coli* 23S rRNA
is also supported by the log-probability calculations (Fig. 6d, Supple-
mentary Fig. 12a). The models also support sequences with U to C
mutations at nearby positions 2473 and 2474, either individually
(U2474C) or in combination, and predict these to confer thermo-
tolerance (Supplementary Data 5, Fig. 6d), consistent with their slight
enrichment in hyperthermophilic 23S rRNAs in GARNET (Supplemen-
tary Data 4, Supplementary Data 1). The sequences generated by the
GNN and RNA LMs often introduced compensatory base pair changes
in H89, and the models yielded lower ΔΔlog*P* values when only one
nucleotide in a pair was changed (Supplementary Fig. 12b). However,
we did not prioritize base pair changes in the H89 stem, as compen-
satory base pairs were deemed unlikely to have a dramatic impact on
ribosome stability at the initial stages of unfolding based on our

previous work[47]. Given the importance of H89 late in ribosome
assembly, we made mutations at positions 2473, 2474, and 2477 to test
their effects on *E. coli* 50S subunit thermostability. We also re-
examined the A loop mutations in the closing loop of H92 at positions
2554 and 2555 (Supplementary Data 5, Fig. 6c, f, g). As noted above,
U2554C and U2555C mutations in H92 (H92-CC) were previously
shown to globally stabilize the *E. coli* 50S subunit[47].

We purified in vivo assembled 50S subunits with U2473C-U2474C,
U2477C, U2554C-U2555C, and U2477C-U2554C-U2555C mutations
using MS2-tagging[50,51]. We additionally purified WT *E. coli* 50S subunits
with an MS2-tag to serve as a control. To test for thermal stability, we
pre-incubated the 50S subunits at 65 °C, cooled them to room tem-
perature, and then assessed if they maintained activity after heat
treatment in an in vitro translation reaction (Fig. 6e). We found that
H89 mutations U2473C-U2474C and U2477C do not affect the activity
of ribosomes at 37 °C (Fig. 6f). However after pre-incubation at 65 °C,
50S subunits with a U2477C mutation are roughly twice as active as WT
subunits (Fig. 6f), indicating that this mutation stabilizes the 50S
subunit. By contrast, ribosomes with the U2473C-U2474C mutations
are not more active than WT after pre-incubation at 65 °C (Fig. 6f),
indicating these mutations do not stabilize the 50S subunit in this
assay. We also examined whether the stabilization from mutations in
H89 and H92 are additive. 50S subunits with U2554C-U2555C (H92-CC)

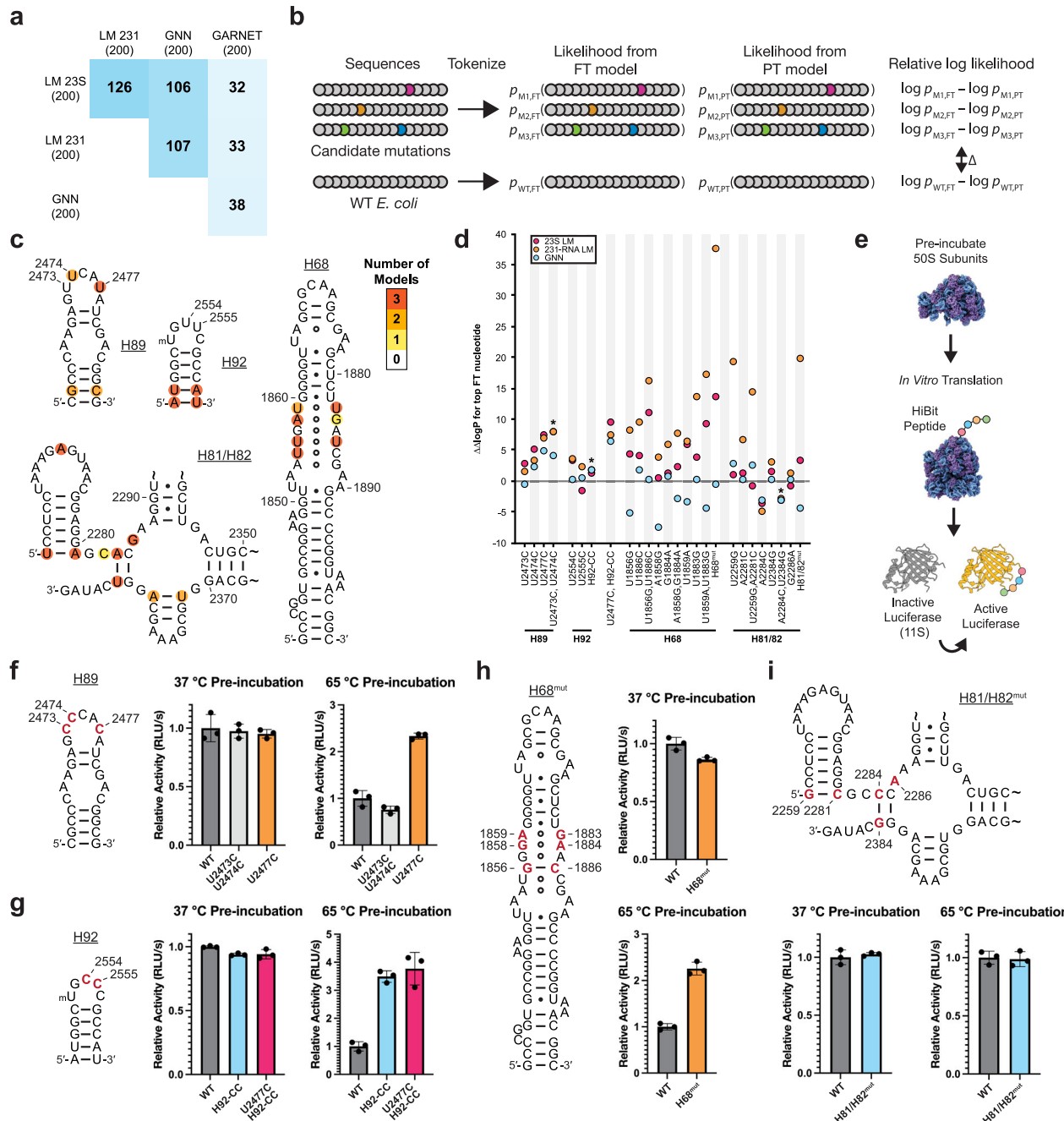

**Fig. 6 | Mutations in 23S rRNA predicted by deep learning models to confer thermostability on *E. coli* ribosomes. a** Matrix showing the overlap in the 200 positions with highest Jensen-Shannon divergence in finetuned (FT) model-generated versus pretrained (PT) model-generated sequences for the GNN, LM models, and in the hyperthermophilic versus total GARNET 23S rRNA sequences. **b** Strategy for calculating ΔΔlog*P* values for candidate mutations, using log likelihoods of sequence generation from FT versus PT models, with WT *E. coli* serving as a normalization control. **c** Positions within four regions of the *E. coli* 23S rRNA with JSD values ranked in the top 200. Coloring indicates the number of models which identify each position. **d** Analysis of the four regions in panel (**c**) for candidate thermostabilizing mutations. For each position, the most frequent nucleotide in FT-generated sequences (top FT nucleotide) is grafted into the *E. coli* 23S rRNA sequence and used to calculate ΔΔlog*P* for the 23S LM, 231-RNA LM, and GNN

models. Overlapping values are denoted with an asterisk in the graph for clarity. **e** Schematic for the heat-treatment in vitro translation assay. Purified 50S subunits are incubated at the indicated temperature, cooled to room temperature, and then added to a HiBiT in vitro translation assay. The peptide complements an inactive protein fragment to form an active luciferase. **f–i** Activity of pre-incubated ribosomes in the HiBiT in vitro translation assay. Secondary structures of helices H89 (**f**), H92 (**g**), H68 (**h**), and H81/H82 (**i**) of *E. coli* 23S rRNA. Positions that were mutated in this study are shown in red. For panels **f** through **i**, WT, and mutant 50S subunits all contain an MS2 tag (Methods). Relative activity is calculated as the slope of the initial increase in luminescence during translation and normalized to the WT value at the given temperature. Data and error bars represent the average and standard deviation of three reactions, respectively. Source data for panels (**f**) through (**i**) are provided as a Source Data file.

mutations were more than threefold as active as WT subunits after pre-incubation at 65 °C. Addition of the U2477C mutation to the H92-CC mutations (U2477C-H92-CC) did not increase the stability past that of the H92-CC mutations on their own (Fig. 6g).

We focused on two additional regions in domains IV and V—helix H68 and helices H81 and H82—that contained multiple positions ranking in the top 200 highest JSD values (Fig. 6c, Supplementary Figs. 9–11). Domain IV is the penultimate domain to fold[31,48] and

includes helix H68, which harbors multiple non-canonical base pairs in the middle of the stem[42] with high JSD values. Notably with the GPT-like LMs, pairwise changes for all three of these non-canonical pairs independently resulted in higher log probabilities than changes to individual bases in a pair (Fig. 6d). Due to the fact that H68 includes multiple adjacent non-canonical base pairs, we tested the H68 changes as a group (H68^mut) and found that changing these base pairs increased *E. coli* ribosome stability (Fig. 6h). A second set of nucleotides with high JSD values that cluster in helices H81/H82 involve four changes in two Watson-Crick-Franklin base pairs, from A-U to C-G pairs (Fig. 6i). The fifth nucleotide, G2286, is unpaired (Fig. 6i) and interacts with ribosomal protein bL33. In contrast to H68, the two base pair changes do not result in more positive ΔΔlog*P* values using any of the language models, and the change G2286A gives differing results across the three language models (Fig. 6d). As with H68, we tested these five mutations as a group (H81/H82^mut), given their close proximity in the structure. Consistent with our hypothesis that canonical base pair changes are unlikely to be limiting for thermostability in the *E. coli* context, the H81/H82 mutations result in ribosomes with equivalent activity and thermostability as WT ribosomes (Fig. 6i). Thus, taking the H68 and H81/H82 mutation groups together with the mutations in H89 and H92, the GNN and LM models are able to inform 23S rRNA mutations that stabilize the *E. coli* 50S subunit in four of six cases tested here, and one tested previously[47].

## Discussion

Here we show that two distinct deep learning frameworks, a GNN and a generative RNA LM, could be used to identify functional RNA mutations in the ribosome. RNA structure prediction and design using deep learning has lagged behind efforts for proteins, in large part due to the limited abundance and quality of available RNA sequence and structural information[13]. While it might be possible in the future to refine the RNAcentral database, this will require careful handling of RNA sequence duplication, as well as rigorous division of sequences into training and validation sets. To address the database problem, we created GARNET, an entirely new RNA database built from the GTDB[36]. The GTDB incorporates not only bacteria and archaea that can be grown in the lab, but also genomes for uncultured microbes, expanding the scope of RNA sequence alignments that can be obtained for bacteria and archaea. The GTDB framework also enables linking phenotypes to genomes, as well as multiple sequences from the same genome across alignments, which can aid studies of protein and RNA complexes. We used a machine learning approach[40] to assign an optimal growth temperature to each reference genome in the GTDB, building on experimental measurements[38,39]. We then tested whether these temperatures, assigned to the RNAs identified in the GTDB genomes, could be used to identify thermophilic mutations that stabilize the *E. coli* ribosome. Using two different deep learning architectures–a graph neural network (GNN) and an RNA language model (LM)–we were able to identify mutations in *E. coli* 23S rRNA that stabilize the 50S subunit to heat treatment (Fig. 6). Importantly, instead of relying on generated sequences individually, we generated sets of 1000 sequences to analyze, in order to avoid possible issues with model-generated artifacts (i.e. "hallucination"). We used two different kinds of sequence interrogation to identify stabilizing mutations, namely Jensen-Shannon divergence (JSD) and model probability calculations (Fig. 6). Sorting positions by JSD identifies individual positions that differ the most between the pretrained and finetuned generated sequences. Calculating the model probabilities allowed us to evaluate whether these mutations are still supported when grafted individually into the *E. coli* 23S rRNA sequence. We focused on identifying individual mutations, or at most several substitutions. Future work to mine the combinatorial effects of multiple mutations, as well as higher-throughput assays, may help maximize the ability to query the GNN and RNA LMs for stabilizing RNA mutations. Overall, the methods used here to identify functional

RNA mutations, by comparing models pretrained on the entire GARNET RNA dataset to models finetuned on GARNET hyperthermophilic sequences, could likely be adapted for protein engineering.

Thermostabilizing mutations identified using GNNs and RNA LMs are distinct from those that could be gleaned through direct analysis of natural 23S rRNA sequences in GARNET, consistent with these deep learning models extracting new information from the sequence data. This may be in part due to sequence co-dependence. For example, a nucleotide change at U2477C is strongly predicted to confer higher thermostability in the *E. coli* context using the JSD calculation and model probabilities, and mutations U2473C-U2474C have a higher probability of conferring thermostability relative to the WT *E. coli* sequence. However, only U2477C is capable of stabilizing the *E. coli* 50S subunit in the in vitro translation assay used here (Fig. 6f), suggesting positions 2473 and 2474 may have other dependencies. In the *E. coli* ribosome, the U2477 base stacks with A2476 and interacts with an arginine side chain of ribosomal protein bL36 (Supplementary Fig. 13a). Cytosine has a larger dipole moment than uridine[52], which could increase the strength of the rRNA-ribosomal protein interaction and thereby stabilize the *E. coli* 50S ribosomal subunit. The predicted H89 mutation maintains and potentially strengthens this rRNA-ribosomal protein interaction despite the RNA LMs having no knowledge of ribosomal proteins. By contrast U2473C-U2474C mutations showed no improvement in ribosome stability, although U2473 contacts an arginine side chain in ribosome assembly factor ObgE during 50S subunit maturation[49]. Notably, ribosome assembly factors are missing from the in vitro assay we used here, suggesting the U2473C mutation might still be beneficial in the assembly of destabilized engineered ribosomes in vivo.

While a GNN utilizes both sequence and structural information for training, GPT-like LMs use only sequences for training. Interestingly, the use of a structural component in the GNN model allowed these models to perform as well as the GPT-like LM with an order of magnitude fewer parameters (Fig. 5, Supplementary Data 3). Notably, we identified a unique feature of RNA that favors representation of overlapping nucleotide triplets as tokens for training GPT-like LMs. These tokens outperform other embedding schemes by substantial margins in our tests. It is possible that this representation captures a fundamental property of RNA, in which nucleotide base stacking is the dominant driving force for RNA structural stability[53]. This contrasts with proteins, in which higher-order structure depends more on backbone features, i.e. backbone hydrogen bonding in secondary structure elements. Tokenization of nucleotides as overlapping triplets effectively represents each of the 4 nucleotides 16 different ways, with additional representations for beginning and ending tokens. The fact that overlapping triplet encoding substantially decreases the perplexity of the resulting LMs suggests that these different representations of the 4 nucleotides capture distinct features that are hard to train in a simpler token scheme. In principle the embedding dimension for nucleotides encoded individually could be increased and might possibly capture this information. For example with proteins, single-amino acid encoding results in "clustering" of amino acids by physicochemical properties[54]. However, for nucleotides this is likely infeasible due to the fact that model parameters and memory use scale as the square of the embedding dimension for a transformer-based model[55]. Projecting the total embedding dimensions of overlapping triplets to single nucleotides would likely require a model with over 100-fold more parameters and memory than used here. While we were unable to find sets of parameters and hyperparameters that allowed training of the RNA LMs using single-nucleotide tokens, this could be further explored in the future. Nevertheless triplet-encoding with a 1-nucleotide shift should serve as a useful approach for the relatively small models we developed here.

Protein language models can serve as a foundation for structure prediction. For example, the ability of ESM-2 to predict correct amino acids in a sequence, as measured by a decrease in the model perplexity, correlates strongly with the ability of the model to serve as a basis for

protein tertiary structure prediction[2]. For RNA, in which alternative secondary and tertiary structures may play important functional biological roles[56,57], starting from an RNA language model may prove crucial for success in future structural prediction efforts. RNA LMs could also benefit from coupling to additional data. For example, combining the RNA LM with a protein LM could help refine searches for mutations in proteins that confer thermostability to the ribosome, i.e. in ribosomal proteins or maturation factors. Language models for RNA could also benefit from information on nucleotide modifications. These modifications can have profound effects on nucleotide contributions to RNA secondary and tertiary structure, and hence RNA function. However, information on nucleotide modifications is scarce apart from a very small select number of organisms. Future efforts to expand post-transcriptional modification databases could help improve deep-learning approaches for RNA. RNA language models could also be combined with experimental data, for example chemical probing data as a means of introducing additional structural information into the model. Finally, there is also room to expand RNA sequences in the GARNET database, which could improve the RNA LMs created here. For example, the larger diversity present in the SILVA 16S rRNA database, which includes ribosomal RNA sequences from species without a sequenced genome, suggests the GTDB could grow in species clusters by many times in the coming years. The GTDB also presently lacks eukaryotic, mitochondrial, and chloroplast genomes, as well as those of viruses. However, even with the above limitations, we show that deep learning models including LMs with optimized triplet encoding can be built and trained using RNA sequences extracted from the GTDB, and applied to RNA functional engineering.

## Methods

### RNA sequence searches and multiple sequence alignment construction

Sequences for the three ribosomal RNAs were identified by searching the corresponding Rfam 14.9 covariance models (23S rRNA: archaea RF02540, bacteria RF02541; 16S rRNA: archaea RF01959, bacteria RF00177; 5S rRNA: RF00001) against genomes in the Genome Taxonomy Database (GTDB) v214.1. The representative genomes of each GTDB species cluster was searched using Infernal 1.1.4 with an e-value cutoff of 1e-5 and omitting hits shorter than 85% of the model length, keeping the most significant hit per genome. If no such hit could be found, then any available non-representative genomes for that species cluster was searched, in order of increasing CheckM contamination, which is provided in the GTDB metadata.

For each ribosomal RNA family, multiple sequence alignments were created by aligning the Infernal hits to a single Rfam covariance model (23S rRNA: RF02541; 16S rRNA: RF00177; 5S rRNA: RF00001). The alignments were further filtered for quality by 1) removing sequences with >5% ambiguity characters, 2) removing sequences that aligned to <85% of the Rfam consensus positions, 3) removing sequences with a length greater than two standard deviations above the mean (greater than one standard deviation for 16S and 23S rRNA), and 4) removing sequences with a fraction of non-canonical base pairs (not Watson-Crick-Franklin or G-U pairs) in the Rfam consensus secondary structure greater than two standard deviations above the mean to remove potential pseudogenes. For the GNN approach, the 23S rRNA alignment was further processed to remove positions that aligned to insertions relative to the Rfam RF02541 model and positions that are not present in the *E. coli* 23S rRNA sequence from PDB 7K00.

For the expanded 228-RNA dataset, we selected 256 Rfam families that are present in bacteria and archaea, contain 10 or more sequences in the Rfam seed, and have 100 or more consensus positions. The models were then searched against each GTDB species representative genome using Infernal 1.1.4[20] with an e-value cutoff of 1e-5, allowing multiple hits per genome. Across all models, hits with any overlapping nucleotides were resolved by keeping the hit with the lower e-value.

The resulting sequences were then aligned to their respective Rfam covariance model. These alignments were filtered for quality in the same way as described above for rRNA sequences, except sequences that aligned to <90% of Rfam consensus positions were removed. Rfam families with fewer than 10 sequences after filtering were excluded from further analysis, resulting in 228 RNA families in the final dataset.

To compare alignment diversity relative to existing RNA alignments, each alignment was filtered at a range of fractional identity cutoffs using a greedy algorithm implemented by two methods: VSEARCH v2.15.2[58] with options --cluster_fast --iddef 0 --id  and esl-weight (HMMER version 3.4)[59] with options --rna -f --idf . VSEARCH takes unaligned sequences as input, while esl-weight requires input sequences to be aligned. For 23S rRNA, the comparison database was SILVA 138.1 LSURef NR99, and for 16S rRNA, SILVA 138.1 SSURef NR99[18]. The full-length SILVA sequences were aligned using SINA 1.7.2[60] to the corresponding ARB file for esl-weight comparisons. For 5S rRNA, two comparison databases were used: 5SRNAdb[37] and Rfam 14.9[17] full alignment for RF00001. 5SRNAdb provides aligned sequences and Rfam sequences were aligned using Infernal to the Rfam covariance model RF00001. For the TPP riboswitch, cobalamin riboswitch, and T-box leader RNA, the comparison databases were Rfam 14.9 full alignment for RF00059, RF00174, and RF00230, respectively, aligned using Infernal to the corresponding covariance model.

### Generation of RNA training and test sets for training deep learning models

We applied hierarchical clustering with CD-HIT-EST[61] to generate training and test sets from 231 Rfam RNA families extracted from the GTDB genomes. To increase cluster diversity, CD-HIT was customized by reducing cluster_thd to 60% in the cdhit-common.c ++ script (line 358) and recompiling the software. Sequences for each Rfam family were independently clustered at decreasing percent identities as follows: 90% with n-mer = 8, 80% with n-mer = 5, 70% with n-mer=4, and 60% with n-mer=4. While the rRNA families were diverse at the 60% identity level, the remaining Rfam families were generally less so due to the stringent filters used in the Infernal search (described above). We therefore used the following strategy for dividing these Rfam sequences into an overall training and test set. First, for the 124/231 remaining Rfam families that had sufficient sequence diversity at the 60% level, clusters were randomly sorted into the training and test sets until up to 33% of sequences from a family were in the test set. Then, intact Rfam families with single clusters were randomly selected for the test set until the test set contained 10% of the total tokens. Rfam families with intermediate diversity, i.e. that had dominant clusters within them, were kept intact in the training set. For models requiring MSA format, sequences were then formatted using esl-reformat (HMMER version 3.4)[59]. The same method was used to split the 5S, 16S and 23S datasets, except 5% of sequences were reserved for testing.

### Growth temperature curation and prediction

Optimal growth temperatures (OGTs) were predicted by TOME[40] from proteome sequences from each representative genome in the Genome Taxonomy Database (release 214.1), yielding a dataset of 85,205 OGTs. This compares to a total of 13,011 out of the 85,205 GTDB species with an OGT listed in TEMPURA and/or GOSHA databases. To determine the accuracy of the TOME predictions, the $R^2$ value was calculated against the optimal growth temperatures from the TEMPURA (Release 200617)[38] and Gosha databases (accessed on 23 October 2023)[39], for all species absent from TOME's training set (Fig. 2a. $n = 7404$ and 3346 species for Gosha and TEMPURA, respectively). OGTs from TOME were further validated by inspecting the NCBI Isolation Source of species in the GTDB metadata. Isolation sources indicating direct acquisition from environments warmer than 60 °C were categorized as "hyperthermophilic," while the remaining

isolates were classified as "not hyperthermophilic" (Supplementary Data 2). Manual labels were compared to classifications based on TOME, where species with a predicted OGT ≥ 60 °C were categorized as hyperthermophiles (Fig. 2c).

### Structural analysis of 23S rRNA for graph representations

The Graph Neural Network (GNN) model takes as input information on the nucleotides' structurally proximal neighbors, represented by a graph. Further, the GNN model is trained on the Infernal MSA of 23S rRNA sequences from GARNET, truncated to align with the *E.coli* 23S rRNA sequence in PDB entry 7K00 and tailored to the Rfam RF02541 model, as detailed in the '**RNA sequence searches and multiple sequence alignment construction**' section. This alignment is further referred to as the 'GNN MSA'. To train the GNN, we generated a graph from a structural distance matrix based on the *E.coli* 23S rRNA structure from PDB entry 7K00, adjusting the nucleotide coordinates in the distance matrix to align with those in the GNN MSA. This was accomplished through a procedure outlined below.

To generate the aligned distance matrices, we chose 18 representative archaeal and bacterial ribosome structures from the PDB (3CC2, 4W2E, 5DM6, 5NGM, 8HKU, 6SKF, 6SPB, 6V39, 7JI1, 7NHK, 7OOD, 7S0S, 7S9U, 7SFR, 8A57, 8FMW, 7K00, and 4YBB), extracting the 23S rRNA chains. Using a custom script, we converted nucleotides with post-translational modifications in these structures to sequences with canonical A, C, G, and U, further referred to as the 'PDB-derived sequences'. We then produced distance matrices by calculating the minimum all-to-all atom distances between nucleotide pairs in the PDB files. To align these distance matrices with the GNN sequence alignments, a multi-step matrix adjustment procedure was implemented (see Supplementary Fig. 2). First, to account for the absence of unstructured regions in the PDB-derived sequences, these were aligned with the corresponding rRNA FASTA sequences from the PDB-derived sequences using MAFFT[62]. Empty rows and columns were inserted into the distance matrices corresponding to the locations of the alignment gaps, signifying the regions of unstructured nucleotides. Subsequently, in the second step, the FASTA sequences of the 18 rRNAs were aligned with the GTDB-derived 23S rRNA sequences using Infernal as outlined above in section '**RNA sequence searches and multiple sequence alignment construction**', ensuring all rRNA sequences were mapped onto a consistent coordinate framework with the necessary gaps and insertions. Empty columns and rows were again positioned at the coordinates of the gaps and insertions in the distance matrices. Finally, in the third step, rows and columns in the distance matrices that correspond to the gaps and insertions specific to the *E.coli* 7K00 sequence in the Infernal MSA, were removed, replicating how the GNN MSA was created. The resulting aligned distance matrices' nucleotide coordinates matched their counterpart coordinates in the GNN MSA.

The aligned distance matrices, showing internucleotide distances in Å, were transformed into binary contact maps, where '1' denotes contact and '0' indicates no contact, with two different methods. In the first intuitive method, contact '1' was assigned to pairs of nucleotides having a distance below a certain distance cutoff. Analysis of the contact map alignment involved summing the 18 maps (see top-right halves of the plots in Supplementary Fig. 3a-d and Fig. 3e). The alignment's accuracy was confirmed by the precise matching of secondary structures across the maps. To quantitatively assess rRNA structural homology, we introduced a structural correlation metric

$$Corr\left(A_{ij}, B_{ij}\right) = \frac{\sum_{i,j} A_{ij} B_{ij}}{\sqrt{\sum_{i,j} A_{ij}^2 \sum_{i,j} B_{ij}^2}}$$

where $A_{ij} = A_{ij}(d)$ and $B_{ij} = B_{ij}(d)$ are the two contact maps compared at a distance cutoff $d$, with $i,j$ being the nucleotides coordinates. Pairwise

correlation of the contact maps generated at $d = 12$Å was on average 0.94 and 0.95 for bacterial species and for archaeal species, respectively, and 0.88-0.90 when comparing bacterial to archaeal 23S rRNA, which indicated that most of the structural features were identical between all ribosomal subunits, and prompted us to combine the sequences for training the GNN (see Supplementary Fig. 3e). We further analyzed the average correlation between the contact maps as a function of distance cutoff $d$ (see Supplementary Fig. 3f), and saw that the structural correlation did not significantly improve for $d$ above 12 Å.

In the second method, similar to the one applied in the original *Structured Transformer* model introduced by Ingraham et al[41]., we sorted pairs of nucleotides according to their distances, and selected the $k$-nearest neighbors for each. To justify the use of the second method, we analyzed the distributions of internucleotide distances returned for different amounts of nearest neighbors $k$ (see Fig. 3d). We observed that by choosing a certain $k$, all internucleotide distances below a certain cutoff are included (e.g. ~12 Å for $k = 50$). We further saw high similarity of the $k$-nearest neighbors contact maps with the contact maps generated for the corresponding cutoff distances captured by a given $k$ nearest neighbor value (see Fig. 3e and Supplementary Fig. 3a–d). We concluded that the two methods for generating contact maps could be used interchangeably, and we chose to proceed with the $k$-nearest neighbors method for generating the graph and training the GNN model.

### GNN model

As described above, the Graph Neural Network (GNN) RNA model takes as input a contact map describing the 3D fold of the RNA family that is being modeled to construct a fixed graph. Each node in the graph corresponds to a conserved position of the RNA family MSA. Each node is connected to the $k$-nearest neighbors. The graph contains node and edge features. Node features consist of a learned absolute positional encoding with 16 hidden features as well as information about the sequence. As in Ingraham et al[41]., this sequence information is causally masked during the decoding process. The edge features consist of the sinusoidal relative positional encodings and the pairwise distance between nodes in the graph use 16 radial basis functions spaced between 0 and 20 Ångstroms, as previously described[41]. All node and edge features were mapped to a hidden dimension of 128 with a learned linear layer. The model leverages the transformer encoder-decoder architecture of Ingraham et al[41]. A single encoder layer and three decoder layers were used. All sequences were tokenized using one token per nucleotide with an alphabet consisting of the four nucleotides (A, U, C, and G) as well as a gap character (-) and an "unknown" character (X). The "unknown" character is found in sequences where, due to sequencing issues, the identity of the nucleotide was not determined.

For training, we performed a sweep on the 23S pretraining set, varying both $k$-NN ($k$-nearest neighbors on which to perform message-passing), and layer dimension. We trained across values of $k = \{5, 10, 20, 50, 100\}$ and layer dimension = $\{64, 128\}$ with learning rate 1e-3, to profile the contribution of added structural context and/or dimension on autoregressive perplexity. We trained all models using a dropout rate of 10% and a label smoothing rate of 10%. For training, we initially randomly partitioned 20% of the training set into a validation set, to allow early stopping based on validation perplexity for the hyperparameter sweep. We found that structural context begins to saturate after $k = 50$ nearest neighbors. Using the best set of hyperparameters on the holdout, divergent test set ($k = 50$, layer dimension = 128), we then partitioned the 10% of the training set for validation for early-stopping on the final pretrained model. We pause training after validation perplexity stopped improving for 5 epochs, training the model for 32 epochs. For finetuning on hyperthermophilic sequences, we lowered the learning rate to 1e-4, and finetuned the pretrained model

($k = 50$, layer dimension = 128). We similarly held out 10% of the hyperthermophiles training set as a validation set to allow early stopping based on validation perplexity. Finally, we measured the performance of the model by calculating test perplexity after training for 50 epochs. Extended details for each model are available in Supplementary Data 3.

## RNA language model pretraining and finetuning on hyperthermophilic sequences

To construct a Generative Pretrained Transformer (GPT) RNA language model, RNA sequences were converted to n-gram tokens of 1, 2, or 3 nucleotides, with a step size of one between tokens (Fig. 4a). A small GPT model, nanoGPT[43], was then adapted to train an RNA language model for comparisons of these token schemes, using 23S rRNA sequences from GARNET. Batch sequences were adjusted to be aligned at index = 0, and used padding if the sequence included the RNA 3′-end. Padding tokens were excluded from loss calculations. We were unable to find suitable hyperparameters for training models with single-nucleotide embeddings. For hyperparameter optimization, we divided the 23S rRNA training set described above into training and validation sets using CD-HIT-EST[61] for hierarchical clustering (85 M/ 4 M tokens in the training/validation sets). Final models were trained using the full training and tests sets for 23S rRNA described above, using the test set as a validation set (89 M/5.5 M tokens in the training/ validation sets). The architecture and hyperparameters for GPT models in the comparisons shown in Fig. 4b were the following: context window = 384 tokens, attention layers = 18, attention heads = 6, embedding dimension = 300, learning rate = 5e-5 decayed over 100,000 iterations to 5e-6, AdamW optimizer beta2 = 0.998, batch size = 18, use of Flash attention[63], and with the nanoGPT model modified to use rotary positional embeddings (RoPE) for relative positional information[44]. We also replaced layer normalization in the transformer layer blocks with root-mean-square normalization[64]. We also tested the use of non-overlapping dinucleotide and non-overlapping triplet-nucleotide encodings. Non-overlapping dinucleotide encodings could be optimized to some degree, possibly benefitting from multiple representations of each nucleotide in the token set. However, non-overlapping dinucleotides require additional tokens to account for RNAs with an odd number of nucleotides and are not as intuitive to interconvert between tokens and nucleotides. We therefore did not pursue non-overlapping embeddings further.

We trained the final RNA language models using the overlapping triplet-nucleotide scheme (n-gram of 3 with step size 1), and with RoPE applied to each attention layer. The final model for the 231 RNA set similarly used the train/test sets described above, with the test set used for validation (274 M/31 M tokens in the training/validation sets). We used early stopping based on the validation loss score to output the final model checkpoint files. The hyperparameters and perplexity values of the pretrained models are given in Supplementary Data 3. 16S and 23S rRNA sequences were generated from the pretrained 231-RNA LM using 100 nucleotides of *E. coli* 16S or 23S rRNA, respectively, at a generation temperature of 0.2. These sets of 100 sequences were aligned using the MAFFT aligner in Wasabi[65], with the *E. coli* sequence included for comparison purposes in Fig. 4d and e.

RNA language models trained on 23S rRNA sequences from GARNET were finetuned using hyperthermophilic 23S rRNA sequences from GARNET identified as described above. Hyperthermophilic sequences were divided into a training set and validation set splits based on their partitioning in the data used for pretraining, i.e. hyperthermophilic sequences in the training set of the pretraining data were used in the finetuning training set, and hyperthermophilic sequences in the validation set of the pretraining data were used in the finetuning validation set. As with the pretrained models, early stopping based on the validation loss score was used to output the final model checkpoint files. We also finetuned the RNA language model

pretrained on the 231-RNA dataset using a similar workflow (Supplementary Data 3).

## Analysis of 23S rRNA sequences to identify candidate thermophilic mutations

Full-length 23S rRNA sequences were generated from the pretrained and finetuned GNN and LM models using a seed sequence beginning with the 5′ end of *E. coli* 23S rRNA composed of 100 nucleotides (GNN) or 384 nucleotides (LM). Sequences were generated in sets of 1000 using a range of "temperature" scaling factors of the model output logits, then aligned to the consensus 23S sequence using the Rfam covariance model RF02541 (LSU_rRNA_bacteria). The GNN-generated sequences lacked regions in the uL1 and bL12 binding regions, which were missing in PDB entry 7K00. These regions were masked in subsequent analyses. Sequences generated from the LMs aligned to the Rfam model for 23S rRNA across their entire length (Supplementary Fig. 6a, c and Supplementary Fig. 7a, c). By contrast, the GNN models deleted local RNA segments with higher frequency at the higher generation temperatures tested (Supplementary Fig. 5a, c). Although shorter as a function of increasing temperature, the GNN sequences still aligned well to the Rfam model (Supplementary Figs. 5–7).

To choose an appropriate temperature for generating sequences, they were analyzed for their 23S rRNA-like properties as follows. First, generated sequences were scored against the Rfam covariance model RF02541 using cmsearch in the Infernal suite of programs[20]. The cmsearch score is a combination of sequence and secondary structure conservation, giving a global view of the 23S-like properties of the generated RNAs. However, a 1-2% change in secondary structure may not affect the score substantially if the rest of the ~3k long sequence is conserved. We therefore used a second metric, the fraction of consensus base pairs in the RF02541 model aligned to each sequence that deviate from canonical G-C/C-G, A-U/U-A, or G-U/U-G pairs, compared to proportion of base pairs disrupted in natural 23S rRNA sequences from GARNET. Generated sequences were also visually checked for alignment properties using the SILVA Alignment, Classification, and Tree (AC) service[60], together with the SILVA-associated Wasabi sequence viewer[66].

To identify candidate mutations in *E. coli* 23S rRNA that might confer thermostability, we analyzed generated sequences with a generation temperature of T = 0.5 for all GNN and RNA LMs, except for the 231-RNA FT model, where we used a generation temperature of T = 0.3. We first aligned the generated sequences to the Rfam RF02541 covariance model using cmalign in the Infernal suite of programs, and trimmed the alignment to positions corresponding to the *E. coli* 23S rRNA sequence. We calculated the Jensen-Shannon divergence (JSD) at each nucleotide position in the 23S rRNA alignment, comparing nucleotide frequencies for sequences generated by the pretrained models and models finetuned on hyperthermophilic sequences, after masking positions used to seed sequence generation ($n = 100$ for GNN, $n = 386$ for LM to account for tokenization) and with nucleotide occupancy <50%.

Since JSD-based sorting considers each position in the sequence independently, we also used log probability values for candidate 23S sequences to assess mutations. Using the probability of a sequence being generated by an RNA language model allows us to assess whether candidate mutations work in the *E. coli* 23S rRNA context or may depend on other co-occurring mutations, i.e. compensatory mutations in base pairs. Notably, many of the highest-scoring JSD sites do in fact correspond to base paired positions in 23S rRNA, and the deep learning models generated compensatory mutations at both nucleotide positions to maintain the base pair. However, given the large number of mutations in each GNN- and LM-generated sequence, on the order of 200 or more per sequence, it is also possible that candidate mutations might not function in an otherwise WT *E. coli* 23S background.

We used four probability calculations in our log probability analysis (Fig. 6b). First, we calculated the log probability of the finetuned (FT) model generating a mutated *E. coli* 23S rRNA sequence, and compared this to the log probability from the pretrained (PT) model. Second, we calculated the log probability of the FT model generating the wildtype *E. coli* 23S rRNA, and compared this to the log probability from the PT model. We evaluated the mutant log probability difference between FT and PT normalized to wildtype log probability difference as ΔΔlogP. Mutant sequences with a positive ΔΔlogP are supported by the FT model better than the PT model relative to the wildtype *E. coli* 23S rRNA. As controls, we generated all possible single-nucleotide mutations in the *E. coli* 23S rRNA sequence and calculated log probabilities for each of these being generated from the FT or PT models. We found the average difference in log probabilities from the FT and PT models, ΔΔlogP, to be close to 0 (−0.82 for the 23S rRNA GNN, 0.07 for the 23S rRNA LM, and −0.52 for the 23S rRNA LM). Comparing single mutations to this reference distribution allowed us to assess the percentile of individual candidate thermostabilizing mutations. Multiple-mutation cases should be compared to an analogous reference distribution considering all the mutations in a sequence in the probability calculations, which could be used to investigate nucleotide dependencies learned by the models. We chose not to comprehensively assay these due to the computational complexity.

## Cloning and ribosome purification

For ribosome expression, a modified version of the pLK35 plasmid[67], which contains an IPTG inducible tac promoter followed by the 5S, 16S, and 23S rRNA with the MS2-tag from Nissley et al[50]. inserted in helix H98, was used. 23S rRNA mutations were introduced to the pLK35 plasmid using the corresponding primer set (Supplementary Data 6) and the In-Fusion Cloning kit (Takara Bio). All sequences were confirmed with full plasmid nanopore sequencing (Plasmidsaurus and Elim Bio).

MS2-tagged ribosomes were expressed and purified as previously described[47] with adaptations. pLK35 plasmids were transformed into NEB Express I$^q$ cells (NEB) which are a bL21 derivative that constitutively expresses the lac repressor. Transformants were grown overnight in LB media and the following day were diluted 1:100 in 1 L of LB media with 100 μg/mL ampicillin. The cultures were grown at 37 °C with shaking and once the cultures reached an OD$_{600}$ of 0.6, expression of the rRNA was induced with 0.5 mM Isopropyl ß-D-1-thiogalactopyranoside. Induced cultures were grown for three hours at 37 °C and then cells were pelleted and resuspended in 30 mL of buffer A (20 mM Tris−HCl pH 7.5, 100 mM NH$_4$Cl, 10 mM MgCl$_2$) with a Pierce protease inhibitor tablet (Thermo Fisher). The cell suspension was lysed by sonication and the lysate was clarified by centrifugation at 14,000 rpm (34,000 $x$ $g$) for 45 min in a F14-14 × 50cy rotor (ThermoFisher). The clarified lysate was then loaded onto a sucrose cushion with 24 mL of buffer B (20 mM Tris−HCl pH 7.5, 500 mM NH$_4$Cl, 10 mM MgCl$_2$) with 0.5 M sucrose and 17 mL of buffer C (20 mM Tris−HCl pH 7.5, 60 mM NH$_4$Cl, 6 mM MgCl$_2$) with 0.7 M sucrose in Ti-45 tubes (Beckman-Coulter). Ribosomes were pelleted by centrifugation at 27,000 rpm (57,000 $xg$) for 16 h at 4 °C and then resuspended in dissociation buffer (20 mM Tris−HCl pH 7.5, 60 mM NH$_4$Cl, 1 mM MgCl$_2$).

MBP-MS2 fusion protein was purified as previously described[32]. 10 mg of MBP-MS2 protein was loaded onto a MBP Trap column (Cytiva) that was equilibrated with MS2-150 buffer (20 mM HEPES pH 7.5, 150 mM KCl, 1 mM EDTA, 2 mM 2-mercaptoethanol). The column was washed with 5 column volumes (CV) of buffer A-1 (20 mM Tris−HCl pH 7.5, 100 mM NH$_4$Cl, 1 mM MgCl$_2$) and the resuspended ribosome pellet (-100 mg) was then loaded onto the column. The column was washed with 5 CV buffer A-1 followed by 10 CV of buffer A-250 (20 mM Tris−HCl pH 7.5, 250 mM NH$_4$Cl, 1 mM MgCl$_2$) and ribosomes were eluted with 10 mL of elution buffer (20 mM Tris−HCl pH 7.5, 100 mM NH$_4$Cl, 1 mM MgCl$_2$, 10 mM maltose). The 50S subunit sample was then concentrated using a 100 kDa cut-off spin filter (Millipore) and washed

with buffer A-1. 50S ribosomal subunits were quantified using the approximation of 1 A$_{260}$ = 36 nM, flash frozen, and stored at −80°C. WT untagged 30S subunits were purified from *E. coli* MRE600 as previously described[47].

Endogenous *E. coli* 50S subunit contamination was quantified using semi-quantitative RT-PCR. The rRNA from 50 pmol of MS2-purified 50S subunits was denatured at 95 °C and precipitated with 4 M LiCl. 75 ng of rRNA was reverse transcribed and amplified with 8 PCR cycles using the OneStep RT-PCR kit (Qiagen) and primers MS2_quant_F and MS2_quant_R (Supplementary Data 6). DNA products were resolved on a 10% TBE gel, visualized with SYBR gold stain (Thermo Fisher), and quantified using Image J software[68]. Uncropped gel images for Supplementary Fig. 13 are provided in the Source Data.

## HiBit in vitro translation reactions

The 11S nanoluciferase fragment that is complemented by the HiBit peptide to enable luminescence[69] was purified as previously described[47]. In vitro HiBit translation assays were performed as previously described[47] with adaptations. 50S ribosomal subunits were diluted to 1.4 μM in buffer A described above with a final concentration of 10 mM MgCl$_2$. The subunits were then incubated at 37 °C or 65 °C as indicated for 15 minutes, and then cooled at room temperature for 15 minutes. After cooling, an in vitro translation mixture was assembled using the ΔRibosome PURExpress kit (NEB): 3.2 μL solution A (NEB), 1 μL factor mix (NEB), 250 nM pre-incubated 50S ribosomal subunits, 500 nM WT untagged 30S ribosomal subunits, 1 U/μL Murine RNAse inhibitor (NEB), 400 nM 11S NanoLuc protein, 1:50 (v/v) dilution of Nano-Glo substrate (Promega), and 1 ng/μL of DNA template encoding the HiBit peptide[69] (final volume of 8 μL). 2 μL of the in vitro translation mixture was placed in a 384 well plate per well, and luminescence was measured for one hour in a Spark Plate Reader (Tecan) set to 37 °C. Ribosome activity was calculated by determining the slope of the initial linear region of each in vitro translation reaction. The reported ribosome activities are the average from three HiBit in vitro translation reactions.

## Benchmarking of GenerRNA

The GenerRNA model was downloaded from the following GitHub repository: https://github.com/pfnet-research/GenerRNA.git and built following the instructions provided in the README file. The GenerRNA pretrained model, trained to 330,000 iterations on the deduplicated RNAcentral dataset, as provided on Hugging Face, was additionally fine-tuned on two datasets, GARNET-all and GARNET-hyperthermophiles. To do so, each dataset was first reformatted from the default FASTA format to have each sequence on a single line lacking headers, as required by GenerRNA, using a custom Python script. Following this, each dataset was partitioned into training and validation sets and tokenized using the included tokenizer_bpe_1024 scheme and the default vocabulary to ensure consistency with the original GenerRNA training workflow. Two versions of GenerRNA were then fine-tuned on these datasets using the included finetuning example config file and finetune.py script for 50,000 iterations on four A4500 GPUs to ensure the validation loss plateaued.

We then sampled 1000 sequences from each GenerRNA model using the provided sampling.py code. We tested a number of parameters including temperature, seed, token generation strategy, and max tokens to generate the most 23S-like set of 1000 sequences possible from the default GenerRNA model, the GenerRNA model fine-tuned on GARNET-all, and the GenerRNA model fine-tuned on GARNET-hyperthermophiles. Broadly, these parameters ended up being a 100-nucleotide seed from the 5' end of the *E. coli* 23S rRNA, --max_new_tokens 520, --temperature 0.5 or 1.0 for the pretrained and fine-tuned models respectively, and --strategy top_k. We chose 520 tokens, because for the 1024-token vocabulary, sequences are

compressed about 5x once tokenized (Figure S1 in the GenerRNA manuscript[45]). Increasing the number of tokes did not result in longer generated sequences.

In addition to these parameters, we were forced to edit the code of the model.py and sampling.py scripts to suppress the <|endoftext|> token, as without this, sequences would almost always terminate within 100-150 nucleotides of starting. This was accomplished by setting the <|endoftext|> logit value to negative infinity, as well as adding a flag that allowed us to define a forbidden token, <|endoftext|>. However, this did not totally alleviate the early sequence termination issues but did allow us to generate longer, though still prematurely terminated, 23S-like sequences. Following generation, we assessed the sequences using two metrics, cmsearch score and the fraction of non-Watson-Crick base pairs, as described for the RNA LM models. With both metrics, the sequences generated using the default GenerRNA model yielded poor cmsearch score values, more dissimilar to naturally occurring 23S rRNA sequences than those generated by the GARNET RNA LM models. The fraction of non-WatsonCrick base pairs on the surface look 23S-like for the default model. However, this metric does not account for truncation of the sequences at the 3' end. The sequences generated by the GARNET fine-tuned GenerRNA models catastrophically failed, generating only fragmentary sequences with low cmsearch scores and exceptionally high non-Watson-Crick base pair fractions.

### Reporting summary

Further information on research design is available in the Nature Portfolio Reporting Summary linked to this article.

## Data availability

Associated data, models, and the GARNET Database[70] are provided here: https://doi.org/10.5281/zenodo.14003346. PDB coordinates are available at:7K00. 4YBB. 6SPB. 8A57. 7S0S. 7NHK. 7SFR. 8FMW. 5DM6. 7OOD. 6V39. 7JIL. 5NGM. 4W2E. 7S9U. 6SKF. 3CCZ. 8HKU. Source data are provided as a Source Data file. Source data are provided with this paper.

## Code availability

Code described in this work is publicly available on Github (https://github.com/Doudna-lab/GARNET_DL[70]) and at the following https://doi.org/10.5281/zenodo.13999143.

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

## Acknowledgements

This manuscript is the direct outcome of a community effort, generously hosted by the Innovative Genomics Institute and University of California Berkeley Electrical Engineering and Computer Sciences Department. We thank the RNA Consortium for the useful discussions that inspired this work, including: Dr. Benjamin Adler, Professor Adam Arkin, Dr. Jigyasa Arora, Teena Bajaj, Dr. Daniel Bellieny-Rabelo, Dr. David Ding, Dr. Hope Henderson, Aditya Iyer, Ryan Keivanfar, Annabel Large, Kenneth Loi, Dr. Andrew Murdoch, Ruchir Rastogi, Omer Ronen, Adit Shah, Rachel Weissman and Chengzhong Ye. These discussions were coordinated by Dr. Daniel Bellieny-Rabelo, Elaine Zhang, Keana Lucas and Theodore Prompichai. We are also grateful to Jamie Irvine who reviewed the manuscript and provided essential insights. Finally, we thank Dr. Robert Wang, Alex Lotz and Tafadzwa Chimbindi from Amazon Web Services, whose extensive support and enablement made this event possible. This work was supported by the NSF Center for Genetically Encoded Materials (C-GEM, CHE-2002182: J.H.C., Y.S., A.J.N.) and the NSF Graduate Research Fellowships Program (P.Y.). Y.S. is a Don Brown Awardee of the Life Sciences Research Foundation. S.C. is supported by the Masason Foundation and the Mercatus Center's Emergent Ventures Fellowship. P.S. is supported by the Swiss National Science Foundation Mobility fellowship (P500PB_214418). H.S. is an HHMI Fellow of The Jane Coffin Childs Fund for Medical Research. E.E.D. is funded by the CIRM Training Program (EDUC4-12790). J.A.D. is an investigator of the Howard Hughes Medical Institute, and research in the Doudna lab is supported by the Howard Hughes Medical Institute

(HHMI), NIH/NIAID (U54AI170792, U19AI135990, UH3AI150552, U01AI142817), NIH/NINDS (U19NS132303), NSF (2334028), DOE (DE-AC02-05CH11231, 2553571, B656358), Lawrence Livermore National Laboratory, Apple Tree Partners (24180), UCB-Hampton University Summer Program, Mr. Li Ka Shing, Emerson Collective and the Innovative Genomics Institute (IGI).

## Author contributions

J.H.C. conceptualized the RNA LM project. Y.S., M.I.T., P.S., H.N., R.S.B., and H.S. conceptualized strategy for project implementation. Y.S., C.L., and M.I.T. produced the GARNET database. J.H.C., H.N., S.C., R.S.B., and A.M.I. developed the software and methodology for the models. Y.S., J.H.C., C.L., and H.N. validated models. Y.S., C.L., M.I.T., J.P., P.S., and H.S. contributed to bioinformatics analyses. M.I.T, P.Y., E.E.D., J.P., P.S., H.S., and T.P. reviewed taxonomic data. A.J.N. designed and performed all experiments. J.H.C., Y.S., C.L., M.I.T., J.P., P.S., and A.J.N. prepared figures and visual representations. J.H.C., Y.S., C.L., M.I.T., H.N., S.C., P.S., and A.J.N. wrote the initial draft and all authors reviewed and edited the manuscript. J.H.C. and J.A.D. were responsible for funding and supervised the project.

## Competing interests

J.H.C. is the founder, board and SAB member of Initial Therapeutics. The Regents of the University of California have patents issued and pending for CRISPR technologies on which J.A.D. is an inventor. J.A.D. is a cofounder of Azalea Theratupics, Caribou Biosciences, Editas Medicine, Evercrisp, Scribe Therapeutics, Intellia Therapeutics, and Mammoth Biosciences. J.A.D. is a scientific advisory board member at Evercrisp, Caribou Biosciences, Intellia Therapeutics, Scribe Therapeutics, Mammoth Biosciences, The Column Group, and Inari. J.A.D. is Chief Science Advisor to Sixth Street, a Director at Johnson & Johnson, Altos, and Tempus, and has a research project sponsored by Apple Tree Partners. The remaining authors declare no competing interests.
