## [Transparent Peer Review file · Nature Communications]

RNA language models predict mutations that improve RNA function

Corresponding Author: Professor Jamie Cate

Version 0:

Reviewer comments:

Reviewer #1

(Remarks to the Author)

This paper presents the development of generative models for RNA function. It contains a large amount of work by a broad collaborative consortium. It contains three main contributions:

- A new RNA sequence database (GARNET) that allows to perform diverse MSA for downstream model training. This is an excellent resource that will likely be useful to many in the field.
- Two generative models based on a graph neural network and a language model based on the widely adopted GPT architecture.
- Experimental validation of a mutation suggested by both models that improves thermostability of the 23S ribosomal RNA in E coli.

This is a timely contribution on a very active subject - a number of groups have developed similar databases and models for RNA structure/function prediction. My main comments concern the novelty of the approach, comparison with the literature (which is moving at a rapid pace) and streamlining of the presentation of both models.

1. The new database is a very strong contribution that will likely be adopted by many groups.
2. The paper presents two models, and it is unclear what the caveats and advantages of each are. Particularly, both models suggest the same mutation to improve thermostability of the 23S rRNA in E coli, and it would be good to expand on the comparison between the two. I also note that the GNN model is based heavily on Ingraham et al, and it is unclear what novelty their architecture brings to the literature. The GPT model, on the other hand, has a bespoke architecture and the authors went to great lengths to study its performance; the finding the overlapping 3-mer tokenization is a valuable technical observation that may help other groups in this space.
3. Benchmark with other models: the paper does not compare predictions against other similar models, and in my opinion this is a key limitation of this work. Other models include RNA-FM (Chen et al. 2022), Evo (Nguyen et al. 2024), and Ribonanza (He et al. 2024); the latter seems particularly amenable for a head-to-head comparison with the presented model. These works should also be cited in this manuscript.
4. Presentation: At various points the paper tends to compare results with protein structure/function predictors (AlphaFold, ESM), but I think this is unnecessary and somewhat makes the contribution look more incremental than what it really is. RNA structure/function prediction is a very active subject and a standalone topic, with fundamental differences from the protein prediction tasks, as the authors point out in various places.
5. Experimental validation: the paper presents a solid validation and the text contains several interesting comments about compensatory or silent mutations found by the model; it would be very valuable to explore this further through the lens of epistasis and compare/quantify predicted epistatic interactions using some of the methods in the literature, for example

comparing with the additive model or other more advanced approaches (eg Otwinowski et al 2018). In principle, one could use their RNA model to generate a map of local and global epistatic interactions, and this could be a valuable demonstration of the utility of the generative models.

(Remarks on code availability)

Reviewer #2

(Remarks to the Author)

The authors propose a new RNA sequence database called GARNET (Gtdb Acquired RNA with Environmental Temperatures) that is built by searching for RNAs in the genomes available in the GTDB (Genome Taxonomy Database). GARNET offers a much greater diversity of sequences for almost all RNA families compared to state-of-the-art datasets. Additionally, GTDB species were mapped to optimal growth temperatures (OGT) from existing growth temperature databases. Since these databases contain experimental OGTs for only a fraction of the GTDB species, missing values were inferred with the popular OGT prediction model TOME.

Secondly, GARNET database was used to train two types of generative models: structure-aware GNN and GPT language model. Models were trained on next-token prediction task and were first pretrained with available RNA sequences and then finetuned with only hyperthermophilic 23S rRNA sequences. Nucleotide distributions of pretrained and finetuned models were compared to detect mutations that would increase the thermostability of the RNA. Notably, this approach was used to detect Escherichia coli 50S subunit mutations, which stabilize the ribosome.

This work has value for the field of RNA research. For a very long time, the application of machine learning methods in the field has been hampered by the lack of diverse high-quality RNA data. I am sure that the diverse dataset provided in this study will alleviate this problem and enable the development of better foundation models.

Furthermore, protein foundation models have been successfully used to detect stable mutations, but no such thing (at least to my knowledge) has been done for RNA molecules until now. I support the use of RNA generative models. From a machine learning perspective, the contribution of this work is modest since the models are largely adaptations of those previously developed for proteins.

The work supports its claims with experimental methods. More precisely, the study experimentally shows that mutations detected by the developed models indeed increase the stability of the E. coli 50S subunit.

Investigating whether the 2477 mutation also occurs in the H89 region of hyperthermophilic sequences used for fine-tuning would be beneficial.

Additionally, the authors need to explain the rationale behind presenting results for various generation temperatures of the 231-RNA PT model when generating 16S and 23S sequences (0.2), and analyse mutations (0.5).

If I understood everything correctly, the schematic of the sequence decoder in figure 3a contains an error. Only nucleotides between the referent nucleotide and the 3' end should be connected with the referent nucleotide with the blue arrow. However, three nucleotides between the referent nucleotide and the 5' end are connected to it using the blue arrow as well. This is wrong because of the causal masking in the decoder (the model should be able to see "past" parts of the sequence).

It is important to include validation perplexity curves for the single nucleotide tokenization in Figure 4b. Given that most RNA language models have employed single nucleotide tokenization, any insights into the benefits of using triplets with a stride of 1 would be valuable.

Furthermore, it is unclear why the model utilizing the single nucleotide tokenization strategy was trained for only 100,000 iterations while all other models underwent significantly longer training periods, as detailed in Supplementary Table 3.

Another thing that would be interesting to see is whether the other available RNA autoregressive language models such as GenerRNA [1] can be used for mutation detection. Such models are pre-trained on different datasets (mostly RNACentral) and their performance would indicate whether the GARNET database is especially beneficial in comparison to other datasets.

The authors should clarify the advantage of using the GARNET database in comparison to RNACentral in the context of 23S rRNA.

It would be beneficial for readers to compare developed models on the generation of various RNA sequences, at least 23S rRNA. Which model the authors recommend and why.

I wasn't able to view the contents of the supplementary table 2. All other supplementary tables can be viewed just fine so I assume the file got corrupted during the upload.

The methodology employed in this study is robust and well-aligned with the research objectives. Diversity of the GARNET database has been thoroughly compared with other datasets and potential mutation positions predicted by generative models were compared to ones obtained from sequences in GARNET that acted as a baseline.

The study provides good insight into the content of the proposed datasets and the hyperparameters of the models.

In the manuscript I could not find the explanation how the secondary structure for generated structures was calculated in the context of non canonical pairs.

One thing I would like to see, however, is the duration of model training procedures (how many hours/days was each of the deep learning models trained?) and information about used hardware (GPUs). This would give readers a rough idea of what amount of computational power is needed to reproduce results or train models of similar scale.

Additionally, I was not able to locate weights of pretrained and finetuned models. It is true that training scripts are available but it would be very convenient for many researchers if they could easily access already trained model weights so that they could easily use them for their downstream tasks.

-- References --

[1] Zhao, Y., Oono, K., Takizawa, H., & Kotera, M. (2024). GenerRNA: A generative pre-trained language model for de novo RNA design. bioRxiv.

(Remarks on code availability)

Provided code repository contains everything needed to reproduce the results presented in the study. For example, you can easily reproduce model training experiments by simply running the corresponding script. The same goes for data preparation pipelines.

Code repository provides a README file with clear instructions on how to install code dependencies. The file also contains the description of contents of the directories. I managed to install and run the code with no problems.

Reviewer #3

(Remarks to the Author)

(Remarks on code availability)

Reviewer #4

(Remarks to the Author)

This is a very interesting paper providing valuable insights into developing RNA deep learning models from biologists. While protein language models have gone a long way, there has been only a handful of RNA models and this is a very interesting research topic. The exploration on various tokenization schemes can be quite insightful. Finding the right tokenization is usually the very first step to build any language model, and this is something worth highlighting and exploring. Technical analysis involving the cmsearch scores and mutation analysis are very inspiring. Given this is a new field, I am positive that this kind of work will have an impact.

However, from a machine learning point of view the paper has obvious weaknesses. Please see my comment below. To claim a new RNA language model, the training, evaluation and benchmarks have to be done in very rigorous ways. The authors didn't cite or compare with any of the existing RNA language model, making it impossible to gauge the contribution. This paper would be a better fit as a work interrogating the use of RNA language model for mutation prediction. I suggest that authors consider revising the paper as an analysis paper, rather than a language model paper

****Model presentation, rationale and training details need to be improved****

1. To present new ML models, it is critical to highlight key metrics such as size of the model, size/dimension of the training data, total number of tokens in the data, size/dim of the data for fine-tuning. These information are missing or hard-to-find in the manuscript.
2. RNA seqs have various lengths, and it would be necessary to provide some info about the seq length distribution and how to handle various length in the GNN and GPT models. In particular, GNNs usually take fixed-dimensional input. How did the authors modify the GNN to handle variable-length structures?
3. nanoGPT is a model tailored to natural languages with a large set of possible tokens. It is quite surprising that one can apply almost the same architecture directly to RNA sequences. Any thought on this?
4. The GNN model is poorly represented. Please specify what is the training task, what is the model input and output. Given that RNA 3D structural data are quite limited, is the training dataset very small? Or did you do data augmentation? Based on

the writing, the model seems to be generating RNA sequences based on the input structure? Such a task is not a typical sequence generation task. Instead, it should belong to structure-to-sequence inversion.

5. What is the relation between the GNN model and the nanoGPT model? The paper seems to be stitching two independent projects together. It actually would be very interesting to compare these two models and assess the benefits of incorporating structural information for RNA generation.

****Validation of the trained models and quality of generated sequences are insufficient****

For text, image and video generation by ChatGPT, DALLE, Sora, humans can easily eyeball the generated outcomes and tell if they are good and resemble nature ones. However, this is impossible with RNA sequences. The paper made some attempts to evaluate the quality of generated sequences (via perplexity, cmsearch and assays), however they are not sufficiently convincing. Language models can hallucinate a lot and there isn't any good way to quantify hallucination, thus

1. The best perplexity value obtained by the model is around ~1.7. This value would be pretty good for a chatbot because natural languages have a huge token space. (For example GPT-2 has a perplexity value of ~2). However, this value ~1.7 is very high in the context of RNA when there are only 4 tokens. While the models have reached the lowest test perplexity via training, it only means that this particular model has been fit to model the distribution of training dataset, but it doesn't mean that the generated sequences are real RNAs.
2. Figure 5 compares the cmsearch scores of generated RNAs compared to nature ones. The plots seem to showcase substantial distribution shift between the generated distribution and nature one. Does this distribution shift mean that the generated sequences are far from nature ones? Figure 6 also indicates that predicted mutation positions are quite different from natural ones. It is hard to tell whether the predicted ones are real or due to hallucinations. I strongly suggest the authors to consider additional evaluation metrics to verify the "naturalness" of generated sequence.
3. It is mentioned that "RNA LM generated some sequences that harbored long stretches of repetitive sequence, resulting in low cmsearch scores". This observation is usually a big red flag that indicates poor training and even bugs in the program.
4. For generating single mutation, how exactly is the language model masked? GPT models work in an autoregressive way, and unlike BERT models, masking is usually added to mask the entire subsequence starting at one position to the end. Can the authors elaborate more on the masking for single nucleotide mutation?
5. A couple of mutation positions were identified from the language model and validated? Is this result cherry-picked? We know that language models can generate very random sequences, and it can predict many possible mutation positions out of the 2900 possibilities. Statistically, it can happen randomly that top predictions overlap with some known hits. Have the authors conducted a thorough analysis/screens of all top-ranked mutations and validated them? This would require doing some statistical test

****Lacks survey of existing RNA language models****

1. There are many works on DNA and RNA language models in various contexts, for examples uni-RNA, RNA-FM, UTR-LM, Eco, RNABert, etc. Some are preprints and some have already been published in Nature/Science series journals. Have the authors tried comparing their model with any existing RNA foundation model?
2. For a paper with the title "RNA language model xxxx", it is expected to provide a comprehensive survey and benchmark against existing pre-trained models.

Other technical questions:

1. The paper seems to be using "validation set" and "test set" interchangeably. Figure 3 uses "test perplexity" and Figure 4 uses "validation perplexity". Do they mean the same thing? Note that validation set should be disjoint from the test set in training any deep learning model.
2. Figure 4b compares perplexity values for different tokenization. Why are they comparable, given that different tokenization schemes corresponds to different size of token vocabulary.
3. In the training curve plots, what as the x-axis label "iterations" mean? The typical metric should be # training epochs. Here iterations are on the scale of 10ks so they cannot be epochs. If they mean number of optimization updates, they are also related to batch sizes which are not thoroughly discussed.

(Remarks on code availability)

Version 1:

Reviewer comments:

Reviewer #1

(Remarks to the Author)

The main update to the revision is the inclusion of new experiments that validate the thermostabilizing effect of additional mutations. This is commendable, but unfortunately the most critical concern from reviewers remain unaddressed.

Specifically, the revision does not compare the proposed models with the predictions of other RNA models, which casts doubt on the methodological contributions of this work. While this is clearly not a machine learning or LLM paper, due to the fast pace of this field it has become standard to compare with at least one other model. The rebuttal rightly argues that some other models (such as Evo) are either hard or impossible to retrain due to lack of details in the sources. However, there are quite a few other models suggested by reviewers (uni-RNA, RNA-FM, UTR-LM, RNABert, etc) where such re-training seems feasible. The rebuttal argues that this is challenging because of the lack of temperature data for all sequences RNA Central data, but there are several workarounds to this, e.g.: a) filter out RNA central sequences for which temperature data is available, even at the cost of reducing the size of training data, and/or b) retrain other models on their own GARNET data to compare model predictions, and/or c) fine-tune the pre-trained models with the temperature data from GARNET, and compare the predicted mutations against their own.

The authors have adequately addressed several technical comments by Reviewer 4, including clarifications on the modelling approaches and details on model architectures. However, their most critical comment, i.e. the lack of comparison with other RNA models, remains unaddressed. Without such comparisons the utility of their models (GNN and nanoGPT) cannot be established. For example, one could fine-tune some models from the literature with the temperature labels from the GARNET database to predict thermostabilizing mutations, and compare these with the GNN and nanoGPT predictions. It is already surprising that both models tend to predict similar mutations (Fig 6d), and perhaps this is a warning sign that other existing models may be able to readily predict those too. If this is the case, the real novelty and contribution of this work would be the GARNET database, which was highlighted as a great resource by all referees, and not the generative models themselves.

(Remarks on code availability)

Reviewer #2

(Remarks to the Author)

The authors have properly addressed many of our objects and comments. However, two open points still require better understanding, and both are related to RNA language models.

1. Pretraining the model on a larger dataset (i.e. RNA central)

Authors responded: "Unfortunately, it is not feasible to assign temperatures to the sequences in RNACentral, given the heterogeneity of the database." (RNACentral comparison)

This is true, but that shouldn't make comparisons with other models impossible, as temperatures are not needed during the pre-training. One could take the model pre-trained with the RNACentral dataset and then fine-tune it with hyperthermophilic 23S rRNA sequences from GARNET. Such an experiment would give a rough idea of how much more beneficial the pre-training with GARNET is compared to RNACentral.

2. The authors in the manuscript stated: "we found that models trained using tokens representing three nucleotides, with a 1-nucleotide shift per token, performed substantially better than using either individual nucleotides or paired nucleotides".

In their response to our comment, the authors answered: "We note that we ran the initial tests of all tokenization schemes for 100,000 iterations. However, since we didn't see any meaningful movement of the validation perplexity for single-nucleotide tokens, we didn't extend those training runs for longer."

Interestingly, triple-tokenization outperformed single-nucleotide tokenization by such a large margin because, in both cases, the next-token prediction should boil down to a next-nucleotide prediction (because of the one nt stride). This gap in the performance is not very intuitive especially that almost all state-of-the-art LLMs such as RiNALMo (Penic et al, 2024) and RNA-FM (Chen et al, 2022) use single nucleotide tokenization. We argue that the finding is of high interest to the scientific community and it is worth exploring. At the moment, the above-mentioned statement is not supported by enough data.

(Remarks on code availability)

Reviewer #3

(Remarks to the Author)

(Remarks on code availability)

Version 2:

Reviewer comments:

Reviewer #1

(Remarks to the Author)

Authors have satisfactorily addressed my comments.

(Remarks on code availability)

Reviewer #2

(Remarks to the Author)

After reviewing the authors' response, we still have several concerns. While we recognize that this paper is not primarily focused on machine learning (ML) or large language models (LLMs), we maintain that all claims should be supported with data to avoid misleading the research community.

#1

The authors addressed Reviewer 1's concern by comparing their model's performance with another autoregressive RNA language model, GenerRNA, which was fine-tuned with the GARNET-all and GARNET-hyperthermophiles datasets. Despite fine-tuning, GenerRNA was unable to generate functional 23S sequences and frequently produced sequences that were too short. Notably, even the fine-tuned versions of GenerRNA struggled to generate 23S-like sequences, despite showing stable validation loss curves and achieving low validation losses when fine-tuned with hyperthermophiles. This comparison offers valuable insights into the performance challenges faced by these models. However, the authors have chosen not to include this comparison in their manuscript, explaining that the GenerRNA paper did not reference nanoGPT. While it would be equitable for GenerRNA to cite nanoGPT, the fact that nanoGPT is licensed under the MIT license means that the lack of citation should not preclude the inclusion of this comparison in the manuscript. Integrating this comparison would significantly enhance the manuscript's contributions to machine learning.

#2

In our previous feedback, we requested that the authors pretrain their language model on RNA central to validate the utility of the GARNET database. Instead, they pretrained GenerRNA. Given the small size of their model, this process should not require extensive resources.

#3

Lastly, we previously highlighted a strong claim made in the manuscript: "Models trained using tokens representing three nucleotides, with a 1-nucleotide shift per token, performed substantially better than those using either individual nucleotides or paired nucleotides." This claim requires robust support with data. We do not expect the authors to develop a BERT-like model for this purpose. However, if they cannot achieve satisfactory performance with single-nucleotide tokens, they cannot assert superior performance of their proposed method without further evidence. Challenges such as suboptimal parameter settings, the need for more parameters, or issues in their code could be influencing their results. These possibilities underscore why detailed ablation studies, parameter examinations, and training with various random seeds are essential in the ML community. We recommend that the authors either support this claim with additional data or moderate the claim's strength.

(Remarks on code availability)

Reviewer #3

(Remarks to the Author)

(Remarks on code availability)

Reviewer #1 (Remarks to the Author)

This paper presents the development of generative models for RNA function. It contains a large amount of work by a broad collaborative consortium. It contains three main contributions:

- A new RNA sequence database (GARNET) that allows to perform diverse MSA for downstream model training. This is an excellent resource that will likely be useful to many in the field.
- Two generative models based on a graph neural network and a language model based on the widely adopted GPT architecture.
- Experimental validation of a mutation suggested by both models that improves thermostability of the 23S ribosomal RNA in E coli.

This is a timely contribution on a very active subject - a number of groups have developed similar databases and models for RNA structure/function prediction. My main comments concern the novelty of the approach, comparison with the literature (which is moving at a rapid pace) and streamlining of the presentation of both models.

We thank the reviewer for their positive views of the work we present in the manuscript.

1. The new database is a very strong contribution that will likely be adopted by many groups.

We thank the reviewer for their view of GARNET.

2. The paper presents two models, and it is unclear what the caveats and advantages of each are. Particularly, both models suggest the same mutation to improve thermostability of the 23S rRNA in E coli, and it would be good to expand on the comparison between the two. I also note that the GNN model is based heavily on Ingraham et al, and it is unclear what novelty their architecture brings to the literature. The GPT model, on the other hand, has a bespoke architecture and the authors went to great lengths to study its performance; the finding the overlapping 3-mer tokenization is a valuable technical observation that may help other groups in this space.

We thank the reviewer for this feedback on the two models. We attempted to address some of these points in the Discussion section, and will clarify those sections of the manuscript. With respect to the GNN model, although Ingraham et al. developed the approach for proteins, RNA is an entirely different polymer. We therefore wanted to test how the Ingraham et al. architecture could perform on the best-defined RNA system presently available, 23S rRNA. First, we were able to define boundary conditions for hyperparameters that are distinct from those that would be used for proteins (See **Fig. 3c**). We also note that the GNN could be much smaller than the GPT model and retain its performance for predicting functional mutations. An important distinction between the GNN and GPT approach is that the GPT approach does not rely on any

structural information and therefore can be applied to any RNA family that has sufficient sequence information. On the other hand, the GNN model is limited to RNA families that have structural information.

3. Benchmark with other models: the paper does not compare predictions against other similar models, and in my opinion this is a key limitation of this work. Other models include RNA-FM (Chen et al. 2022), Evo (Nguyen et al. 2024), and Ribonanza (He et al. 2024); the latter seems particularly amenable for a head-to-head comparison with the presented model. These works should also be cited in this manuscript.

We thank the reviewer for pointing out these models, one of which we cited, and the other two we can add. We debated internally how much we would focus on benchmarking our models against others, and decided to focus more on the functional validation, which we think is more helpful in understanding the value of the models.

For all of these models, the big challenge is setting up pretraining and finetuning, especially with the need to segregate hyperthermophilic sequences. RNA-FM is pretrained on RNA Central, which cannot be parsed to segregate hyperthermophilic sequences due to the database heterogeneity. Furthermore, it would be challenging to avoid train-test leakage for zero-shot evaluations using RNA-FM and GARNET sequences, given partial overlap in sequences in the datasets. Evo is a very large model that took the authors substantial effort to train, and how they did so is not presently available. Furthermore, given the computing resources and memory management issues required to use it, the authors are presently not supporting finetuning of the model. Finally, we think Ribonanza is an exciting development in that it uses chemical probing data for training. We cited the de Lajarte et al. preprint from the Rouskin lab, and should have added Ribonanza which has some conceptual overlap. In terms of benchmarking Ribonanza in comparison to our models, we run into the same challenge of segregating hyperthermophilic sequences for finetuning. We think adapting Ribonanza, which is aimed more at RNA secondary and tertiary structure prediction (Fig. 4 in their preprint), would be better suited to a future study. We have added a statement in the Discussion describing the possible benefit of integrating chemical probing data into future models.

We also have added citations to AlphaFold3 and RoseTTAFold-AllAtom, which can model RNA structure to some degree.

4. Presentation: At various points the paper tends to compare results with protein structure/function predictors (AlphaFold, ESM), but I think this is unnecessary and somewhat makes the contribution look more incremental than what it really is. RNA structure/function prediction is a very active subject and a standalone topic, with fundamental differences from the protein prediction tasks, as the authors point out in various places.

We agree that the RNA structure/function field is a very active field with its own sets of challenges. In light of the reviewer's comment, we have abbreviated discussion of the protein deep learning field, but we do think we need to mention it in certain contexts.

5. Experimental validation: the paper presents a solid validation and the text contains several interesting comments about compensatory or silent mutations found by the model; it would be very valuable to explore this further through the lens of epistasis and compare/quantify predicted epistatic interactions using some of the methods in the literature, for example comparing with the additive model or other more advanced approaches (eg Otwinowski et al 2018). In principle, one could use their RNA model to generate a map of local and global epistatic interactions, and this could be a valuable demonstration of the utility of the generative models.

We thank the reviewer for pointing out Otwinowski et al. 2018 (PNAS). Their work builds a statistical framework for developing models from deep mutational scanning experiments. Because we are limited in the amount of experimental data we have with the present assay, it is not clear how we could develop statistically meaningful global or local epistasis models for our data. We would need on the order of 10^3 - 10^5 or more mutations. This paper is motivating to try to come up with new higher-throughput assays in the future!

We have, however, tested additional clusters of mutations in 23S rRNA based on JSD values (helix H68 and helices H81/H82). One of these clusters also increases the thermostability of the ribosome, whereas the other—consisting primarily of canonical Watson-Crick-Franklin base pairs—does not. We have added these data to the figures (**Figure 6, ED Figures 8-11**).

Reviewer #2 (Remarks to the Author)

The authors propose a new RNA sequence database called GARNET (Gtdb Acquired RNA with Environmental Temperatures) that is built by searching for RNAs in the genomes available in the GTDB (Genome Taxonomy Database). GARNET offers a much greater diversity of sequences for almost all RNA families compared to state-of-the-art datasets. Additionally, GTDB species were mapped to optimal growth temperatures (OGT) from existing growth temperature databases. Since these databases contain experimental OGTs for only a fraction of the GTDB species, missing values were inferred with the popular OGT prediction model TOME.

Secondly, GARNET database was used to train two types of generative models: structure-aware GNN and GPT language model. Models were trained on next-token prediction task and were first pretrained with available RNA sequences and then finetuned with only hyperthermophilic 23S rRNA sequences. Nucleotide distributions of pretrained and finetuned models were compared to detect mutations that would increase the thermostability of the RNA. Notably, this approach was used to detect Escherichia coli 50S subunit mutations, which stabilize the ribosome.

This work has value for the field of RNA research. For a very long time, the application of machine learning methods in the field has been hampered by the lack of diverse high-quality RNA data. I am sure that the diverse dataset provided in this study will alleviate this problem and enable the development of better foundation models.

We thank the reviewer for their positive view of the value of GARNET for RNA research.

Furthermore, protein foundation models have been successfully used to detect stable mutations, but no such thing (at least to my knowledge) has been done for RNA molecules until now. I support the use of RNA generative models.

From a machine learning perspective, the contribution of this work is modest since the models are largely adaptations of those previously developed for proteins.

The work supports its claims with experimental methods. More precisely, the study experimentally shows that mutations detected by the developed models indeed increase the stability of the E. coli 50S subunit.

We thank the reviewer for their interest in the experimental results we obtained using predictions from the machine learning models. We agree some aspects of the machine learning approaches we use are adaptations of architectures developed for proteins (i.e. Ingraham et al.). However, given that RNA is an entirely different polymer than proteins, we think our contributions for the GNN model, in terms of RNA-optimal hyperparameters and model size, are important contributions. We also note that the RNA tokenization using overlapping triples is distinct from how protein sequences are tokenized, and has not been used by others.

Investigating whether the 2477 mutation also occurs in the H89 region of hyperthermophilic sequences used for fine-tuning would be beneficial.

We thank the reviewer for this question. This information is available in the last tab of Supplementary Table 4 ("GARNET" tab). 2477C is present in 35% of all sequences, and 70% of hyperthermophiles. However, this gives a fairly low Jensen-Shannon Distribution score (ranking at position 1007 out of 2904 nucleotides). The ranking is noted on p. 22 in the revised text.

Additionally, the authors need to explain the rationale behind presenting results for various generation temperatures of the 231-RNA PT model when generating 16S and 23S sequences (0.2), and analyse mutations (0.5).

We thank the reviewer for this question. In the case where we compared generating 16S and 23S sequences, we wanted to query the model with a minimal number of nucleotides (100 nts). However, later when we focus entirely on 23S rRNA, we use a larger seed sequence (384 nts), which allowed us to go to higher temperatures, i.e. to sample the distributions more aggressively, for our downstream mutational analyses.

If I understood everything correctly, the schematic of the sequence decoder in figure 3a contains an error. Only nucleotides between the referent nucleotide and the 3' end should be connected with the referent nucleotide with the blue arrow. However, three nucleotides between the referent nucleotide and the 5' end are connected to it using the blue arrow as well. This is

wrong because of the causal masking in the decoder (the model should be able to see "past" parts of the sequence).

We thank the reviewer for catching this mistake. We have updated Figure 3a accordingly.

It is important to include validation perplexity curves for the single nucleotide tokenization in Figure 4b. Given that most RNA language models have employed single nucleotide tokenization, any insights into the benefits of using triplets with a stride of 1 would be valuable. Furthermore, it is unclear why the model utilizing the single nucleotide tokenization strategy was trained for only 100,000 iterations while all other models underwent significantly longer training periods, as detailed in Supplementary Table 3.

We thank the reviewer for this suggestion. Since the single-nt validation perplexity is given in **Supplementary Table 3**, we have not added it to the figure, to keep it easier to visualize. We have now added a call-out to **Supplementary Table 3** where we mention single-nucleotide tokenization. We note that we ran the initial tests of all tokenization schemes for 100,000 iterations. However, since we didn't see any meaningful movement of the validation perplexity for single-nucleotide tokens, we didn't extend those training runs for longer.

Another thing that would be interesting to see is whether the other available RNA autoregressive language models such as GenerRNA [1] can be used for mutation detection. Such models are pre-trained on different datasets (mostly RNACentral) and their performance would indicate whether the GARNET database is especially beneficial in comparison to other datasets.

The authors should clarify the advantage of using the GARNET database in comparison to RNACentral in the context of 23s rRNA.

We thank the reviewer for this idea. Unfortunately, it is not feasible to assign temperatures to the sequences in RNACentral, given the heterogeneity of the database. This is one of the distinct advantages of GARNET, which we note in the Discussion.

It would be beneficial for readers to compare developed models on the generation of various RNA sequences, at least 23S rRNA. Which model the authors recommend and why.

This is an interesting point that intersects with comments by reviewer 1. We had debated internally whether to spend the time benchmarking other models, or to focus on experimental validation. We came to the conclusion that it would be better to use experimental validation in this paper (which we have expanded as describe above and below). We think our main contribution is to highlight that the important considerations for any of the models is the choice of training data and the choice of tokenization.

I wasn't able to view the contents of the supplementary table 2. All other supplementary tables can be viewed just fine so I assume the file got corrupted during the upload.

Thanks for pointing this out. We had gzipped that file to make it smaller. We will upload an uncompressed version in the revision.

The methodology employed in this study is robust and well-aligned with the research objectives. Diversity of the GARNET database has been thoroughly compared with other datasets and potential mutation positions predicted by generative models were compared to ones obtained from sequences in GARNET that acted as a baseline.

The study provides good insight into the content of the proposed datasets and the hyperparameters of the models.

We thank the reviewers for these points of positive feedback.

In the manuscript I could not find the explanation how the secondary structure for generated structures was calculated in the context of non canonical pairs.

Thanks for pointing this out. The secondary structures in **Figure 5** are derived from the consensus secondary structure for Rfam RF02541. We define any changes away from a canonical Watson-Crick-Franklin base pair (i.e. a disrupted WCF base pair) to be “non-canonical” in this context. We clarify this point in the legend to **Figure 5**.

One thing I would like to see, however, is the duration of model training procedures (how many hours/days was each of the deep learning models trained?) and information about used hardware (GPUs). This would give readers a rough idea of what amount of computational power is needed to reproduce results or train models of similar scale.

This is a very good point. One thing that is encouraging about these models is that they are not large, in terms of trainable parameters. The LMs are approximately 20 million parameters, for example (See **Supplementary Table 3**). We trained the LMs on AWS using a p3.2xlarge instance with an NVIDIA Tesla V100 GPU (16GB VRAM). The longest training took less than a week to complete on a single GPU. The GNNs vary in number of parameters, depending on the k-nearest neighbors threshold and layer size, but the final model we used (k = 50, layer_size = 128) had 1,090,054 trainable parameters. For training the final GNNs, we used 4 GeForce RTX 2080's with ~11 GB memory each, enabling data parallelism. This took only ~9 hours for pre-training, and ~8 minutes for fine tuning on the thermophiles.

Additionally, I was not able to locate weights of pretrained and finetuned models. It is true that training scripts are available but it would be very convenient for many researchers if they could easily access already trained model weights so that they could easily use them for their downstream tasks.

We agree these would be valuable to include. They are too large for Github, so we have deposited them in Zenodo at <https://doi.org/10.5281/zenodo.12541208>.

-- References --

[1] Zhao, Y., Oono, K., Takizawa, H., & Kotera, M. (2024). GenerRNA: A generative pre-trained language model for de novo RNA design. bioRxiv.

Reviewer #3 (Remarks to the Author)

We thank reviewer #3 for helping with the peer review of our paper.

Reviewer #4 (Remarks to the Author):

This is a very interesting paper providing valuable insights into developing RNA deep learning models from biologists. While protein language models have gone a long way, there has been only a handful of RNA models and this is a very interesting research topic. The exploration on various tokenization schemes can be quite insightful. Finding the right tokenization is usually the very first step to build any language model, and this is something worth highlighting and exploring. Technical analysis involving the cmsearch scores and mutation analysis are very inspiring. Given this is a new field, I am positive that this kind of work will have an impact.

We thank the reviewer for their positive views of the work.

However, from a machine learning point of view the paper has obvious weaknesses. Please see my comment below. To claim a new RNA language model, the training, evaluation and benchmarks have to be done in very rigorous ways. The authors didn't cite or compare with any of the existing RNA language model, making it impossible to gauge the contribution. This paper would be a better fit as a work interrogating the use of RNA language model for mutation prediction. I suggest that authors consider revising the paper as an analysis paper, rather than a language model paper.

We thank the reviewer for this perspective, and agree this is not a language model paper. In fact, two-thirds of the paper focuses on issues surrounding the use of language models. In the first two figures, we focus on building a more robust RNA training set that can be coupled to phenotypes—GARNET. The language model development constitutes **Figure 3** and **Figure 4**. Even in these figures, we focus primarily on adapting known deep learning architectures to work with RNA. Finally, the analysis of language model output occupies **Figure 5** and **Figure 6**. As noted for reviewer #1, we debated whether to spend our effort benchmarking the models against others, or focus on experimental validation of the models' predictions. Given how little has been done experimentally, we focused our efforts there. At the computational level, we

provide quantitative assessments of the output of the language models, using cmsearch and the consensus secondary structure for 23S rRNA (**Figure 5**). Experimentally, we use the models to identify mutations that make the *E. coli* ribosome more thermostable, in **Figure 6**. We address these in more detail below in response to concerns raised by the reviewer. In the revision, we also include new experiments, as described for reviewer #1 and as described below.

****Model presentation, rational and training details need to be improved****

1. To present new ML models, it is critical to highlight key metrics such as size of the model, size/dimension of the training data, total number of tokens in the data, size/dim of the data for fine-tuning. These information are missing or hard-to-find in the manuscript.

We thank the reviewer for raising these points. All of these aspects were included in the manuscript, but not in one location. We now highlight where these are included for easier reference and consolidate the details in Supplementary Table 3.

2. RNA seqs have various lengths, and it would be necessary to provide some info about the seq length distribution and how to handle various length in the GNN and GPT models. In particular, GNNs usually take fixed-dimensional input. How did the authors modify the GNN to handle variable-length structures?

This is a good question. First for the GNN model, we only used 23S rRNA structures aligned with PDB entry 7k00, and with sequences aligned to Rfam model RF02541. Please see the following two sections in the Methods: **“RNA sequence searches and multiple sequence alignment construction”** and **“Structural Analysis of 23S rRNA for Graph Representations”**. While our GNN uses a single structure to define a sparse attention mechanism and operates on fixed-length inputs, we note that GNN models can be easily adapted to handle variable length inputs. For example, protein inverse folding models are GNN-based generative models that operate on variable length protein structures and sequences.

3. nanoGPT is a model tailed to natural languages with a large set of possible tokens. It is quite surprising that one can apply almost the same architecture directly to RNA sequences. Any thought on this?

To be honest, this was a gratifying result to us. I think it worked well for us for two reasons. First, the code provided by Dr. Karpathy was easy to follow and modify. Second, the overlapping triplet token scheme was critical for using nanoGPT in close to its original architecture (i.e. absolute positional embedding). Rotary embedding helped to an extent with the overlapping paired tokenization, but did not completely level the performance. See **Figure 4B**. Incidentally, it was interesting to see that the next video in Karpathy's LLM series addressed the importance of tokenization for deep learning models, and that this can have a dominant effect independent of model architecture. See: <https://www.youtube.com/watch?v=zduSFxRajkE>

4. The GNN model is poorly represented. Please specify what is the training task, what is the model input and output. Given that RNA 3D structural data are quite limited, is the training dataset very small? Or did you do data augmentation? Based on the writing, the model seems to be generating RNA sequences based on the input structure? Such a task is not a typical sequence generation task. Instead, it should belong to structure-to-sequence inversion.

We thank the reviewer for pointing out the lack of clarity in this section. The GNN model presented here is strictly for 23S rRNA, which is the best represented RNA structure in the PDB and in sequences (apart from 16S rRNA). In principle, it can be used to model any RNA family that (1) has a representative structure in the PDB and (2) the structure is believed to be highly conserved among members of the family. For 23S rRNA is this true as we have a representative structure (PDB: 7K00) and the folded globular nature of the rRNA structure is highly conserved among 23S rRNAs.

As with the GPT model, the GNN is an autoregressive model that generates 23S rRNA sequences. However there are a few important differences. First, the GNN model models fixed length sequences. This is due to the fact that we chose only to model positions of the sequence that could be mapped onto the structure (PDB 7K00). What we get in return, however, is the ability to leverage conserved structural information as an inductive bias for modeling the distribution as 23S rRNA sequences. The PDB structure that the model takes as input is used to derive a k-nearest neighbor graph which the GNN operates on. Whereas the GPT model has no inductive bias about which nucleotides are likely interacting structurally (e.g. base-pairing, stacking etc) the structural model does have this potentially powerful inductive bias. These inductive biases come predominantly from two factors: (1) the sparse graph provides the inductive bias of which positions are likely to be structurally interacting and therefore influencing the identity of other positions, and (2) the edge features in the graph contain distance information that can help the model learn the types of likely interactions between nucleotide positions (e.g. base pairing v.s. stacking).

We note that there is one important difference and possible point of confusion between our structure-conditioned model and those that have been employed elsewhere for protein and RNA modeling. We do not consider our model an inverse folding model. Inverse folding models are trained on many examples of protein (or RNA) and structure pairs and learn a general mapping from structure to sequence. Our model is trained with only a single structure as well as a family of sequences that each share strong structural homology with the representative structure. Our model does not learn a general structure-sequence mapping but rather uses the structural information as an inductive bias to learn the distribution of sequences for this particular family.

5. What is the relation between the GNN model and the nanoGPT model? The paper seems to be stitching two independent projects together. It actually would be very interesting to compare these two models and assess the benefits of incorporating structural information for RNA generation.

This is another good question. First, we wanted to see how adding structure conditioning affected model output, when compared to the nanoGPT output. For comparable performance, the GNN models are on the order of 10x smaller than the nanoGPT-based models, in terms of parameters (see **Fig. 5** and **Supplementary Table 3**). We have expanded on this in the Discussion. Furthermore, the GNN model could be trained using single-nucleotide sequence encoding. Second, we wanted to see if mutations predicted by the GNN model were concordant with the RNA LMs, which gave us more confidence to spend the time making and testing those mutations in the lab (see below for more on the experimental validation).

****Validation of the trained models and quality of generated sequences are insufficient****

For text, image and video generation by ChatGPT, DALLE, Sora, humans can easily eyeball the generated outcomes and tell if they are good and resemble nature ones. However, this is impossible with RNA sequences. The paper made some attempts to evaluate the quality of generated sequences (via perplexity, cmsearch and assays) , however they are not sufficiently convincing. Language models can hallucinate a lot and there isn't any good way to quantify hallucination, thus

Although it is true LMs can hallucinate, we think the reviewer's use of "impossible" is too strong a word, when it comes to analyzing generated RNA sequences. For one, both the GNN and RNA LM architectures identify the same sets of putative thermostabilizing mutations (**Figure 6**, **ED Figures 8-10**). We agree that eyeballing sequences can potentially be misleading, and we wanted to avoid anecdotal cases. This is why we generated sets of 1000 sequences to examine, rather than relying on individual sequences output by the models.

With respect to quantitative analysis, the cmsearch algorithm is state of the art in the RNA field, and is central to identifying RNA family members in Rfam. We added a new dimension to the global analysis provided by cmsearch, by calculating the percentage of disrupted canonical base pairs (i.e. Watson-Crick-Franklin and G-U) in the generated sequences compared to the canonical Rfam model for 23S (**Figure 5e-h** and **ED Figures 6-7**).

1. The best perplexity value obtained by the model is around ~1.7. This value would be pretty good for a chatbot because natural languages have a huge token space. (For example GPT-2 has a perplexity value of ~2). However, this value ~1.7 is very high in the context of RNA when there are only 4 tokens. While the models have reached the lowest test perplexity via training, it only means that this particular model has been fit to model the distribution of training dataset, but it doesn't not mean that the generated sequences are real RNAs.

We thank the author for this question. It's hard to interpret what a given perplexity value means in isolation. This is why we decided to generate sets of 1000 "23S rRNA" sequences to analyze. Otherwise we think we'd be subject to pitfalls of anecdotal analysis. We think that the way we used finetuning for **Figure 5** and **Figure 6** helped us quantify the properties of the generated RNAs. We made the choice of selecting out hyperthermophilic sequences from GARNET to

finetune the model. And as described in more detail below, we made RNA biophysics informed choices on mutations to test.

2. Figure 5 compares the cmsearch scores of generated RNAs compared to nature ones. The plots seem to showcase substantial distribution shift between the generated distribution and nature one. Does this distribution shift mean that the generated sequences are far from nature ones? Figure 6 also indicates that predicted mutation positions are quite different from natural ones. It is hard to tell whether the predicted ones are real or due to hallucinations. I strongly suggest the authors to consider additional evaluation metrics to verify the “naturalness” of generated sequences.

This is an interesting set of questions. For the cmsearch scores at a temperature $T=0.5$, the generated sequences are at least as 23S rRNA-like as the natural distribution of bacteria. The natural archaeal sequences have a lower cmsearch score because we used the Rfam model for bacterial 23S in our analysis (RF02451). Since the cmsearch score reflects both sequence and structure, we used a separate metric focused on preservation of secondary structure, as noted above (i.e. **Figure 5e-h**). From a quantitative perspective, these provide strong evidence that the generated sequences are 23S-like. We continue to think of additional ways of probing the sets of generated sequences, but we think the metrics we present are sufficiently robust to motivate experimental tests of mutations.

3. It is mentioned that “RNA LM generated some sequences that harbored long stretches of repetitive sequence, resulting in low cmsearch scores”. This observation is usually a big red flag that indicates poor training and even bugs in the program.

This is a good point. We typically only saw this pathology when using high temperatures for sequence generation, probably flattening the probability distributions too much. From an operational point of view, the vast majority of the sequences we generated at $T=0.5$ were sufficiently 23S-like (**Figure 5**) that we could use them for downstream analysis in **Figure 6**. This is another reason we generated sets of sequences rather than relying on individual outputs of the model.

4. For generating single mutations, how exactly is the language model masked? GPT models work in an autoregressive way, and unlike BERT models, masking is usually added to mask the entire subsequence starting at one position to the end. Can the authors elaborate more on the masking for single nucleotide mutations?

This is a very good question. We in fact didn’t generate sequences with single mutations. Rather, we generated sets of 1000 sequences (**Figure 5**) and used statistical measures (i.e. the JSD and log probabilities in **Figure 6**) to help choose mutations to test in the lab. We have added a statement about this to the Discussion. And please see below for more on the mutations we chose.

5. A couple of mutation positions were identified from the language model and validated? Is this result cherry-picked? We know that language models can generate very random sequences, and it can predict many possible mutation positions out of the 2900 possibilities. Statistically, it can happen randomly that top predictions overlap with some known hits. Have the authors conducted a thorough analysis/screens of all top-ranked mutations and validated them? This would require doing some statistical test

This is another good question. From an experimental point of view, the assays with which we can work are laborious and at present are not amenable to high-throughput tests. This is why we applied a number of filters before choosing mutations to test in the lab. We describe our method of choosing mutations to test in the “**Analysis of 23S rRNA sequences to identify candidate thermophilic mutations**” section in the Methods. First, we tested a range of temperatures for generating sequences, to be sure we could add enough sequence variation without destroying the 23S-like nature of the sequences (**Figure 5** and **ED Figures 6-7**). Second, we applied a Jensen-Shannon Divergence test to see what positions in the generated thermophilic sequences differ substantially from the sequences generated from the pretrained models (**Figure 6**). However, JSD values consider each position in the sequence independently, and furthermore do not capture the sequence context. For example, we want to know what mutations confer thermostability specifically in *E. coli* 23S. In other words, the JSD value is a useful first filter, but is not necessarily enough to motivate doing an experiment in the lab. We therefore turned to log probabilities where we could computationally test individual mutations, or small sets of mutations, in the context of the *E. coli* 23S background (**Figure 6**). Finally, we used background knowledge of RNA and ribosome biophysics as an additional level of filtering. For example, we decided not to consider mutations to canonical WCF base pairs, as these are less likely to affect the global tertiary fold of the rRNA when ribosomes assemble at 37 °C. RNA tertiary structure typically falls apart at lower temperatures than secondary structure, and we hypothesize that canonical secondary structure changes (i.e. an A-U pair to a C-G) would benefit an organism growing at, say, 80 °C, more than the *E. coli* sequence and ribosomes assembled at 37 °C. We also focused on domains IV and V of 23S rRNA, since these are known to fold last in ribosome assembly.

Since submitting the manuscript, we’ve tested two additional sets of mutations predicted in helix H68, and in the helix H81/H82 region. We tested these as two sets rather than combinatorially. The H68 mutations, which involve a series of non-canonical base pairs, confer stability to the *E. coli* ribosome thermostability. By contrast, the H81/H82 mutations, 4 of 5 involving WCF base paired positions, have WT stability. The H81/H82 result is consistent with our initial decision to focus on non-WCF base paired positions in our experimental tests. We have added these data to **Figure 6** and **ED Figure 11**.

****Lacks survey of existing RNA language models****

1. There are many works on DNA and RNA language models in various contexts, for examples uni-RNA, RNA-FM, UTR-LM, Eco, RNABert, etc. Some are preprints and some have already been published in Nature/Science series journals. Have the authors tried comparing their model with any existing RNA foundation model?

We thank the reviewer for this question, and we will update our reference list to try to keep up with the models that have been released. Given that other models are trained on datasets that cannot be used for our downstream functional analysis (thermostability), we did not explore them. We'd need to retrain them on GARNET, which is beyond the scope of our study given that two-thirds of the paper is focused outside of model training.

2. For a paper with the title "RNA language model xxxx", it is expected to provide a comprehensive survey and benchmark against existing pre-trained models.

We have updated the references to be as comprehensive as possible, but given that none have the ability to associate sequences with temperature as we do in GARNET, we do not think it will be productive to pursue this line of investigation for this manuscript.

Other technical questions:

1. The paper seems to be using "validation set" and "test set" interchangeably. Figure 3 uses "test perplexity" and Figure 4 uses "validation perplexity". Do they mean the same thing? Note that validation set should be disjoint from the test set in training any deep learning model.

This is a good point. We trained the 23S GNN and RNA LMs separately, and took different approaches as a result. The meaning for "test set" and "validation set" is described in the Methods for each model. The "test set" for the GNN model is the same as the "validation set" for the 23S RNA LM. For the RNA LM, we took the approach used for ESM2, where they focused on validation perplexity. In the end, we think the experimental results provide a test of how useful the models are.

2. Figure 4b compares perplexity values for different tokenization. Why are they comparable, given that different tokenization schemes corresponds to different size of token vocabulary.

This is a great question! And it took awhile for us to think this through. In the next token prediction task, the final token of a given sequence actually encodes partial information on the next token. For example, for the overlapping triplets, token N has information on nucleotides 1-2 of token N+1. Thus, all the overlapping token schemes with a 1-nt step effectively have a maximum perplexity of 4.

3. In the training curve plots, what as the x-axis label “iterations” mean? The typical metric should be # training epochs. Here iterations are on the scale of 10ks so they cannot be epochs. If they mean number of optimization updates, they are also related to batch sizes which are not thoroughly discussed.

We thank the reviewer for pointing this out (**Figure 4**). The information on batch size and context window is given in **Supplementary Table 3**. We added this information to the figure legend.

Reviewer #1 (Remarks to the Author)

The submitted revision sidesteps the most critical comments shared by all three reviewers - i.e. benchmarking with other models and further clarifications on the methodological/ML contributions of this work.

While this is clearly not an ML or LLM paper, due to the fast pace of this field it is best practice to compare with at least one other model. It is true that some other models (such as Evo) are either hard or impossible to retrain due to lack of details in the sources, there are few others suggested by reviewers (uni-RNA, RNA-FM, UTR-LM, RNABert, etc) where such re-training is feasible. The response letter mentions that this cannot be done because of the lack of temperature data for all sequences RNA Central data, but there are several complementary workarounds to this: a) filter out RNA central sequences for which temperature data is available, even at the cost of reducing the size of training data, and/or b) retrain other models on their own GARNET data to compare model predictions, and/or c) fine-tune the pretrained models with the temperature data from GARNET, and compare the predicted mutations against their own.

The paper does contain new experiments validating a couple of additional mutations, and this is commendable, but unfortunately the fundamental comments from reviewers remain unaddressed.

We thank the reviewer for these comments. We have worked to find RNA language models comparable to the one we are using here, which is a generative (autoregressive) model. All of the ones mentioned by the reviewer are based on the BERT masked-language architecture, and are not suited to generative tasks. Another one noted by reviewer #2 is RiNALMo, which is also a BERT model (<https://arxiv.org/html/2403.00043v1>). Although there are efforts to rework BERT models for text generation, these are relatively new developments and still lag GPT models by a substantial margin. For example, see Figure 3 in <https://arxiv.org/pdf/2211.15029>.

We did find one generative RNA language model that was recently posted on bioRxiv, GenerRNA (see <https://doi.org/10.1101/2024.02.01.578496> and <https://huggingface.co/pfnet/GenerRNA> or <https://github.com/pfnet-research/GenerRNA.git>). However, the authors do not acknowledge that this model employs nanoGPT as its architecture, and the associated code is copied almost entirely from nanoGPT without attribution. For this reason, we attach our analysis here, rather than putting it in the manuscript. Although not properly citing its sources, we nevertheless tested GenerRNA because it has three features that we think are useful for benchmarking compared to our model. First, it has about 20x the number of parameters compared to our GPT-like RNA LMs. Second, it was pretrained on RNACentral. And third, the authors explored byte-pair encoding (BPE) of nucleotides, rather than defining an encoding scheme directly as we had. The BPE the authors settled on is a 1024-token vocabulary. GenerRNA does not use rotary embedding as we employed in our model.

We benchmarked GenerRNA in three ways, for comparison with our RNA LMs. First, we used the GenerRNA model pretrained on RNACentral sequences to generate 23S rRNA sequences

seeded with the 5'-end of *E. coli* 23S, as done for the RNA LMs. Second, we partitioned GARNET into training and validation sets using the GenerRNA provided code. We fine-tuned GenerRNA using the full GARNET data, followed by 23S rRNA sequence generation. Third, we also partitioned the hyperthermophilic GARNET data into training and validation sets using the GenerRNA provided code. We then fine-tuned GenerRNA using the hyperthermophilic GARNET data, followed by 23S rRNA sequence generation. We then analyzed these sets of 1000 generated sequences as we had done for RNA LM sequences. We looked at the distribution of cmsearch scores, and the fractional non-canonical base pairs, as we had done before. Notably, GenerRNA failed to produce any full-length 23S rRNA sequences from any of the models, whether pretrained or finetuned, in our hands. Other details on how we used GenerRNA are given in the attached Methods below, along with figures showing the results from GenerRNA. From this analysis, we think it is clear that our RNA LMs are superior to the available generative RNA model.

With respect to BERT-based models, we note that masking using overlapping triplets would be non-trivial, as neighboring tokens have information corresponding to the masked token due to the overlapping nature of the embeddings. It is therefore beyond the scope of this manuscript to rewrite the code for these BERT-based models, and to retrain them, to use overlapping triplet encoding.

Reviewer #2 (Remarks to the Author)

The authors have properly addressed many of our objects and comments. However, two open points still require better understanding, and both are related to RNA language models.

1. Pretraining the model on a larger dataset (i.e. RNA central)

Authors responded: "Unfortunately, it is not feasible to assign temperatures to the sequences in RNACentral, given the heterogeneity of the database." (RNACentral comparison)

This is true, but that shouldn't make comparisons with other models impossible, as temperatures are not needed during the pre-training. One could take the model pre-trained with the RNACentral dataset and then fine-tune it with hyperthermophilic 23S rRNA sequences from GARNET. Such an experiment would give a rough idea of how much more beneficial the pre-training with GARNET is compared to RNACentral.

As noted above, we took this approach with GenerRNA, which did not generate full-length 23S rRNA sequences from any of the models: pretrained on RNACentral sequences, finetuned on all GARNET sequences, or finetuned on GARNET hyperthermophiles. This indicates that some feature or features of GenerRNA do not perform well with RNAs of the length of 23S rRNA.

2. The authors in the manuscript stated: "we found that models trained using tokens representing three nucleotides, with a 1-nucleotide shift per token, performed substantially better than using either individual nucleotides or paired nucleotides".

In their response to our comment, the authors answered: “We note that we ran the initial tests of all tokenization schemes for 100,000 iterations. However, since we didn’t see any meaningful movement of the validation perplexity for single-nucleotide tokens, we didn’t extend those training runs for longer.”

Interestingly, triple-tokenization outperformed single-nucleotide tokenization by such a large margin because, in both cases, the next-token prediction should boil down to a next-nucleotide prediction (because of the one nt stride). This gap in the performance is not very intuitive especially that almost all state-of-the-art LLMs such as RiNALMo (Penic et al, 2024) and RNA-FM (Chen et al, 2022) use single nucleotide tokenization. We argue that the finding is of high interest to the scientific community and it is worth exploring. At the moment, the above-mentioned statement is not supported by enough data.

We thank the reviewer for this comment. As we noted above, none of the available RNA language models are generative models, apart from GenerRNA. GenerRNA is also built on the nanoGPT code base (which the authors fail to acknowledge), but does not implement rotary embedding. It does explore the use of byte-pair encoding, settling on a vocabulary of 1024 tokens. However, the resulting models are not capable of generating full-length 23S rRNA sequences in our hands. Furthermore, byte-pair encoding is non-intuitive, compared to the biophysically-inspired triple-nt encoding used in the RNA LMs we report.

As noted above with respect to BERT-based models, masking using overlapping triplets would be non-trivial, as neighboring tokens have information corresponding to the masked token due to the overlapping nature of the embeddings. It is therefore beyond the scope of this manuscript to rewrite the code for these BERT-based models, and to retrain them, to use overlapping triplet encoding.

Reviewer #3 (Remarks to the Author)

Benchmarking of GenerRNA

The GenerRNA model was downloaded from the following GitHub repository: <https://github.com/pfnet-research/GenerRNA.git> and built following the instructions provided in the README file. The GenerRNA pretrained model, trained to 330,000 iterations on the deduplicated RNACentral dataset, provided on Hugging Face, was fine-tuned on two datasets, GARNET-all and GARNET-hyperthermophiles, as described in our manuscript. To do so, each dataset was first reformatted from the default FASTA format to have each sequence on a single line lacking headers, as required by GenerRNA, using a custom Python script. Following this, each dataset was partitioned into training and validation sets and tokenized using the included `tokenizer_bpe_1024` scheme and the default vocabulary to ensure consistency with the original GenerRNA training workflow. Two versions of GenerRNA were then fine-tuned on these datasets using the included finetuning example config file and `finetune.py` script for 50,000 iterations on four A4500 GPUs to ensure the validation loss plateaued (figure panel a).

We then sampled 1000 sequences from each model using the provided `sampling.py` code. We tested a number of parameters including temperature, seed, token generation strategy, and max tokens to generate the most 23S-like set of 1,000 sequences possible for the default GenerRNA model, the GenerRNA model fine-tuned on GARNET-all, and the GenerRNA model fine-tuned on GARNET-hyperthermophiles. Broadly, these parameters ended up being a 100-nucleotide seed from the 5' end of the *E. coli* 23S rRNA, `--max_new_tokens 520`, `--temperature 0.5` or `1.0` for the pretrained and fine-tuned models respectively, and `--strategy top_k`. We chose 520 tokens, because for the 1024-token vocabulary, sequences are compressed about 5x once tokenized (Figure S1 in the GenerRNA manuscript). Increasing the number of tokens did not result in longer generated sequences.

In addition to these parameters, we were forced to edit the code of the `model.py` and `sampling.py` scripts to suppress the `<|endoftext|>` token, as without this, sequences would almost always terminate within 100-150 nucleotides of starting. This was accomplished by setting the `<|endoftext|>` logit value to negative infinity, as well as adding a flag that allowed us to define a forbidden token, `<|endoftext|>`. However, this did not totally alleviate the early sequence termination issues but did allow us to generate longer, though still functionally useless (prematurely terminated), 23S-like sequences (figure panel f). Following generation, we assessed the sequences using two metrics shown in our manuscript: `cmscore` and the fraction of non-Watson-Crick base pairs (figure panels b-e). In both cases, the sequences generated using the default GenerRNA model yielded poor `cmscore` values, more dissimilar to naturally occurring 23S rRNA sequences than those generated by the GARNET RNA LM models (Figure panel b). The fraction of non-WatsonCrick base pairs on the surface look reasonable for the default model (figure panel c). However, this metric does not account for truncation of the sequences at the 3' end. See the attached Figure. The sequences generated by the GARNET fine-tuned GenerRNA models catastrophically failed, generating only fragmentary sequences with low `cmscores` and exceptionally high non-Watson-Crick base pair fractions (figure panels d and e).

Figure. GenerRNA finetuning and benchmarking. **a.** Validation loss values for the GenerRNA pretrained model finetuned on either GARNET-all or GARNET-hyperthermophile sequences. Validation sets were made using the default GenerRNA code. **b-c.** Cmsearch scores (**b**) and fraction of disrupted canonical base pairs (**c**) for sequences generated from the pretrained GenerRNA model. In (**b**) the cmsearch scores for GARNET natural and generated sequences are shown for reference. In (**c**) the fraction of disrupted canonical base pairs (i.e. Watson-Crick-Franklin and G-U) relative to the Rfam RF02541 consensus secondary structure (denoted non-canonical base pairs) in the GenerRNA generated sequences are compared to naturally-occurring 23S rRNAs, and to sequences generated from the GARNET-all RNA LM. **d-e.**

Cmsearch scores (**d**) and fraction of disrupted canonical base pairs (**e**) for sequences generated from the GenerRNA model finetuned on GARNET-all or GARNET-hyperthermophile sequences. **f**. Length of generated sequences from GenerRNA and GARNET RNA LM, compared to the natural distribution of sequence lengths in the GARNET database.

Reviewer #1 (Remarks to the Author):

Authors have satisfactorily addressed my comments.

Reviewer #2 (Remarks to the Author):

After reviewing the authors' response, we still have several concerns. While we recognize that this paper is not primarily focused on machine learning (ML) or large language models (LLMs), we maintain that all claims should be supported with data to avoid misleading the research community.

#1

The authors addressed Reviewer 1's concern by comparing their model's performance with another autoregressive RNA language model, GenerRNA, which was fine-tuned with the GARNET-all and GARNET-hyperthermophiles datasets. Despite fine-tuning, GenerRNA was unable to generate functional 23S sequences and frequently produced sequences that were too short. Notably, even the fine-tuned versions of GenerRNA struggled to generate 23S-like sequences, despite showing stable validation loss curves and achieving low validation losses when fine-tuned with hyperthermophiles. This comparison offers valuable insights into the performance challenges faced by these models. However, the authors have chosen not to include this comparison in their manuscript, explaining that the GenerRNA paper did not reference nanoGPT. While it would be equitable for GenerRNA to cite nanoGPT, the fact that nanoGPT is licensed under the MIT license means that the lack of citation should not preclude the inclusion of this comparison in the manuscript. Integrating this comparison would significantly enhance the manuscript's contributions to machine learning.

We have added our analysis of GenerRNA to the manuscript in the main text, before the section, "**Identifying mutations to stabilize the *E. coli* ribosome**". We also added a **Materials and Methods** section, and added the figure as **Supplementary Figure 8**. We note GenerRNA is built on the nanoGPT code, as well.

With respect to the MIT license of the nanoGPT code, although the code is free to use, it is copyrighted by Andrej Karpathy, so the authorship should be cited. We have done so in our revised manuscript.

#2

In our previous feedback, we requested that the authors pretrain their language model on RNA central to validate the utility of the GARNET database. Instead, they pretrained GenerRNA. Given the small size of their model, this process should not require extensive resources.

We agree training our model should not require extensive resources. However, we ran into severe roadblocks processing RNACentral at the level of preparing the database for training.

Using cd-hit-est, we were only able to complete deduplication after ~19 days (on a single node, with 768 GB RAM, and 64 threads). Furthermore, as is clear from the GenerRNA preprint, it will not be trivial to divide the sequences into training vs. validation sets. The GenerRNA authors simply randomly divided the data, without concern about possible leakage between the training and validation sets. This is highly problematic when it comes to ensuring proper training without overfitting. However, efforts to divide the data into proper training and validation sets is likely to be an involved process and computationally much more expensive than deduplication, based on our experience with RNACentral deduplication and the steps we had to take with the GARNET database. We have added a statement about the possibility of exploring the use of RNACentral in the future, with the need to rigorously deduplicate and divide the data into training and validation sets.

#3

Lastly, we previously highlighted a strong claim made in the manuscript: "Models trained using tokens representing three nucleotides, with a 1-nucleotide shift per token, performed substantially better than those using either individual nucleotides or paired nucleotides." This claim requires robust support with data. We do not expect the authors to develop a BERT-like model for this purpose. However, if they cannot achieve satisfactory performance with single-nucleotide tokens, they cannot assert superior performance of their proposed method without further evidence. Challenges such as suboptimal parameter settings, the need for more parameters, or issues in their code could be influencing their results. These possibilities underscore why detailed ablation studies, parameter examinations, and training with various random seeds are essential in the ML community. We recommend that the authors either support this claim with additional data or moderate the claim's strength.

We have addressed this concern in the Discussion by adding the statement: "While we were unable to find sets of parameters and hyperparameters that allowed training of the RNA LMs using single-nucleotide tokens, this could be further explored in the future. Nevertheless triplet-encoding with a 1-nucleotide shift should serve as a useful approach for the relatively small models we developed here."

Reviewer #3 (Remarks to the Author):

Sequence decoder (autoregressive)

b

3 blue arrows in the red circle should be orange?

Information flow

→ Structure

→ Structure and sequence